# Zero-shot CLIP Class Forgetting via Text-image Space Adaptation

**Alexey Kravets**                                                     *ak3095@bath.ac.uk*
*Department of Computer Science*
*University of Bath*

**Vinay P. Namboodiri**                                               *vpn22@bath.ac.uk*
*Department of Computer Science*
*University of Bath*

**Reviewed on OpenReview:** *https://openreview.net/forum?id=V2SD2uVKEE*

## Abstract

Efficient class forgetting has attracted significant interest due to the high computational cost of retraining models from scratch whenever classes need to be forgotten. This need arises from data privacy regulations, the necessity to remove outdated information, and the possibility to enhance model robustness and security.

In this paper we address class forgetting in vision-language CLIP model. Modern class forgetting methods for CLIP have demonstrated that zero-shot forgetting is achievable by generating synthetic data and fine-tuning both visual and textual encoders with a regularization loss. Our approach shows that class forgetting in CLIP can be accomplished in a zero-shot manner without any visual data by adapting the shared vision-text space of CLIP, thereby making the class forgetting process more efficient. Our method delivers superior results, demonstrating strong performance and complete class removal, regardless of the visual encoder used in CLIP. Furthermore, we explore what exactly is being targeted by the class forgetting algorithm discovering some interesting properties of CLIP features. Full implementation can be found here.

## 1 Introduction

Class removal involves removing specific learned representations (or concepts) from a trained model without full retraining. This process aims to alter the model's understanding of a particular concept while retaining its overall knowledge. In this work, we focus on class removal from the original CLIP model by OpenAI (Radford et al., 2021) that uses either ResNet50 or ViT-B/16 [1] visual encoders, where we aim to remove the associations between certain visual and textual representations in a controlled manner. CLIP is a vision-language model widely used in applications like robotics control (Shridhar et al., 2021), zero-shot object tracking (Solawetz, 2021), and content moderation (Ahmed et al., 2023). As such, it is critical to ensure that CLIP can forget specific concepts when proprietary or sensitive information is involved. For example, if private facial images inadvertently leak into the model during training, CLIP may recognize individuals, raising concerns under GDPR regulations. Such unintended associations could cascade across various applications, posing significant ethical and legal risks.

Class removal can be seen from the perspective of machine unlearning (Xu et al., 2023) which involves removing specific data points from a trained model related to the class to be removed. However, doing this for CLIP is challenging for the following reasons: (a) we do not have access to the original data of the class we want to forget that was used for training CLIP. Thus, any retraining of CLIP to achieve forgetting is not feasible. (b) CLIP is a large parameter model. Even if we did obtain access to the

---

[1]Weights from https://github.com/openai/CLIP/blob/main/clip/clip.py#L30

data that need to be forgotten from CLIP, fine-tuning CLIP would be challenging. To the best of our knowledge, only one work (Kravets & Namboodiri, 2025) has addressed zero-shot unlearning in CLIP. This study demonstrates unlearning in a zero-shot manner without requiring any real data. They indicate that changing weights in both the visual and textual encoders is necessary for class forgetting. In contrast, we show that class forgetting can be achieved without any synthetic or real data, by approaching the problem from the perspective of concept editing where we are not trying to remove the influence of training data but rather break certain associations in the model. Our method modifies only a small part of the textual encoder responsible for projecting the textual representation of the class into the shared image-text embedding space and does not require any synthetic data generation which can be time-consuming making class forgetting process relatively slow.

We recognize that at its core, the contrastive learning for CLIP aims to obtain a joint embedding space for the image and textual representation. Hence, we explicitly use projection of the textual representation to achieve forgetting. Our approach uses a direct optimization of a loss function to modify the text representation projection matrix. While doing so, we need to ascertain the gap between image and text representations is modified only for a select set of classes that we desire to be forgotten while maintaining the gap between image and text representations for the classes that need to be retained. Once we do this for the text representations, we observe that we achieve forgetting for the image-text classes that need to be forgotten and preserve the image-text correspondence for the other classes. After the optimization process is completed, an image for a retained class would still be close to the corresponding text representation. However, for the class that is forgotten, the image representation would be the same as the initial representation but the textual representation would be different as it has been explicitly modified for this class. For optimization we apply an adaptation technique, low-rank adaptation (LoRA) (Hu et al., 2022), to find the minimum change in the text projection matrix optimizing a loss function that ensures that the change is such that the representation of the non-forget classes is retained while altering the representation of the forget class.

We do a **performance** comparison in Tab. 1 showing that our method both outperforms the previous methods and is more robust to different visual encoders achieving perfect class forgetting with ViT (Dosovitskiy et al., 2020) and ResNet (He et al., 2015). We analyse through ablations the importance of the retain and forget **loss components** in Section 7.2 and how **forget class projection place** in the image-text space affects the forgetting ability of the model in Section 7.5. We find that retaining the knowledge of non-forget classes requires the inclusion of semantically similar classes, which can be generated using a large language model (LLM). This is because projecting the forget class to a different space primarily affects the closest classes in the image-text embedding space, thus, it is important to preserve this part of the space, while non-semantically similar classes are retained without explicit inclusion. We conduct a thorough ablation analysis on how the **number of semantically similar classes** affects performance in Section 7.3. Additionally, in Section 7.4 we assess how including semantically different classes affects performance.

We investigate how forgetting occurs. In Section 6.4 we show that there exist some **"magic" neurons** that the forgetting algorithm targets. These weights are such that changing them decreases the dot product for the class to forget the most leading to a change in the class prediction. Furthermore, in the Appendix we show that there is a positive relation between the **difficulty of forgetting a class** and the Frobenious norm in the matrix of weights change. An overview of our approach is provided in Fig 1. Our **contributions** are summarized as follows:

- We improve current state-of-the-art CLIP zero-shot class forgetting as shown in Section 6.1.

- Forgetting is achieved without generating synthetic visual data improving efficiency. We show that no visual data are required to forget a class and textual data are sufficient (Section 4).

- We provide a thorough analysis to understand our method. We show that there exist some "magic" neurons that our method targets to achieve forgetting and why it does that in Section 6.4. We also show that the Frobenius norm correlates with the difficulty of forgetting in the Appendix D.

- A detailed analysis provided in Section 7 validates the choices made in our method and the generalizability of our method.

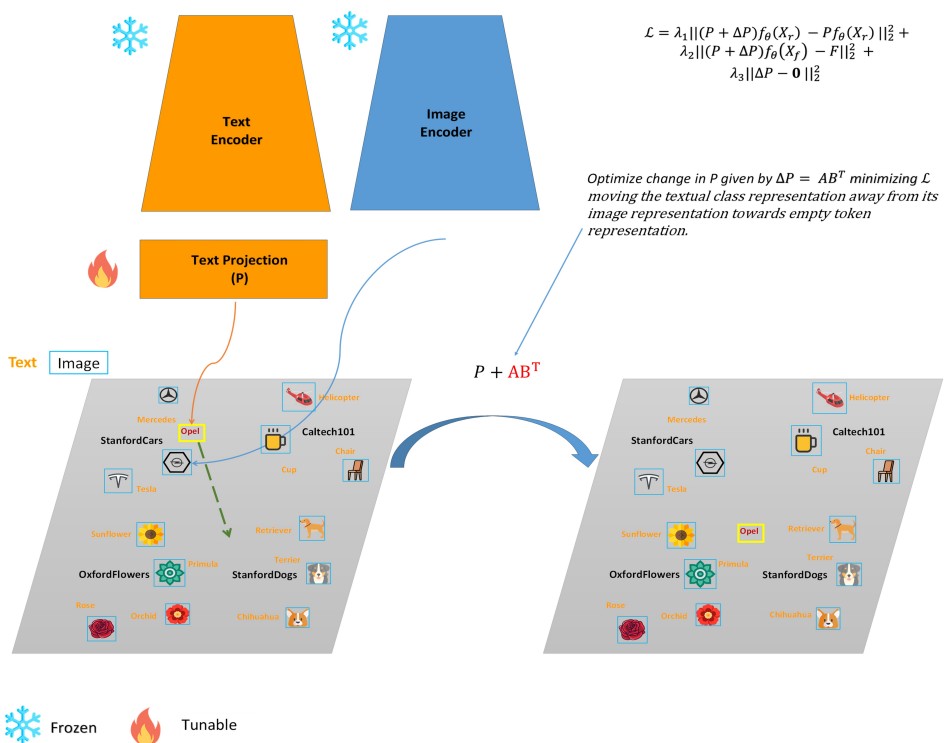

Figure 1: **Overview of the approach.** We utilize LoRA to adapt the projection matrix of the textual representation into the shared image-text space. We ensure the representation for the non forget classes is retained while altering it for the class to forget. To maintain the representation of the non forget classes we generate semantically similar classes using an LLM. The forget class is projected into the empty token representation in the image-text space. In the figure we illustrate the forgetting for the *Opel* class.

## 2 Related Work

**Multi-modal Concept Forgetting**  Li et al. (2024a) propose a method to forget visual recognition of concepts using a single image for multi-modal models. They begin by creating a multifaceted fine-tuning dataset aimed at aligning the forget concept with unseen concepts, assigning it a new visual description, decoupling factual knowledge about it, and preserving unrelated knowledge. The model is then fine-tuned with this data using a dual masked KL-divergence loss. Similarly, Cheng & Amiri (2025) achieve forgetting with a three-term loss function designed to ensure modality decoupling, uni-modal knowledge retention, and multi-modal knowledge retention. Authors of SalUn (Fan et al., 2024) propose to unlearn classification and generation models by computing the weights saliency and updating parameters based on it. These methods require real training examples and are not applicable to the CLIP dual encoder model.

In contrast, Zhang et al. (2024) and Gandikota et al. (2023) achieve class forgetting in the diffusion model by modifying cross-attention mechanisms, thereby disrupting the associations between visual and textual representations of the concepts to be forgotten. Similarly, Gandikota et al. (2024) act on cross-attention in the diffusion model optimizing the key and value matrices mapping target concepts to a new one while preserving some other concepts. This technique allows editing, debiasing and erasure of concepts in the diffusion model. Since CLIP does not utilize cross-attention, these methods are not directly applicable.

To the best of our knowledge, Kravets & Namboodiri (2025) are the first to address zero-shot forgetting in CLIP by applying Lipschitz regularization. This approach guides the embedding of the visual and textual representations of the forget class towards a perturbed embedding, breaking the visual-textual association while retaining knowledge of other classes. They achieve forgetting in a zero-shot manner by generating

synthetic visual data, thus eliminating the need for real examples. The method involves modifying weights in both visual and textual encoders. In contrast, we show that simply updating the projection matrix from the text to image space is sufficient to achieve forgetting still in a zero-shot manner without requiring any real or synthetic images and forgetting more efficiently as synthetic image generation requires time.

**Model Adaptation** Model adapters are task-specific modules added to a pre-trained model to enable it to efficiently adapt to new downstream task without retraining the entire model. Houlsby et al. (2019) inserted sequentially a small multilayer perceptron (MLP) layer between the layers of a pre-trained BERT model while freezing the original pre-trained parameters. Similarly, Chen et al. (2022) added MLP layers but in parallel to the original frozen MLP connecting them in a residual fashion showing superiority compared to the sequential adaptation. Pfeiffer et al. (2021) proposed a unified framework for training and sharing adapters across various tasks. Hu et al. (2022) introduced Low-Rank Adaptation (LoRA) technique which injects trainable low-rank matrices to learn task-specific information without altering the original pre-trained weights significantly. We utilize adapters in a different context, and specifically use LoRA to fine-tune the text projection matrix into the shared image-text embedding space in order to forget a class in CLIP.

## 3 Preliminaries

**CLIP** CLIP is a multi-modal model that understands both visual and textual inputs. It has been trained with a contrastive loss, which helps it learn to represent similar images and their textual descriptions closely in a shared image-text embedding space while keeping dissimilar ones apart. Contrastive loss is defined as:

$$\text{Loss} = -\frac{1}{2}\left(\frac{1}{N}\sum_{i=1}^{N}\log\frac{\exp(\text{sim}(x_i, y_i))}{\sum_j \exp(\text{sim}(x_i, y_j))} + \frac{1}{N}\sum_{i=1}^{N}\log\frac{\exp(\text{sim}(y_i, x_i))}{\sum_j \exp(\text{sim}(y_i, x_j))}\right),\tag{1}$$

where $sim(x_i, y_i)$ is the cosine similarity between an image and text embeddings and $N$ the total number of image-text pairs. Contrastive training enables CLIP to perform various tasks, such as classification and retrieval, in a zero-shot manner.

CLIP employs a dual encoder architecture with separate encoders for images and text. The zero-shot classification process in CLIP operates as follows: the input image is passed through CLIP's image encoder to generate image embedding $T \in \mathbb{R}^d$, that is also normalized, where $d$ is the embedding size. Then, a set of textual descriptions corresponding to the target classes are generated, such as *a photo of a {class}*. These textual class descriptions are converted into text embeddings by CLIP's text encoder, resulting in a classifier weight matrix $W_{preproj} \in \mathbb{R}^{N \times k}$ where $N$ represents the number of classes and $k$ the hidden dimension. Image and text embeddings must live in a shared image-text embedding space so that these can be compared against each other using the cosine similarity where the textual class corresponding to the highest cosine similarity is the predicted class for a given image. For this, $W_{preproj}$ which lives in the text space is projected into the image space via a projection matrix $P \in \mathbb{R}^{k \times d}$ resulting in the final classifier weight matrix $W = W_{preproj}P \in \mathbb{R}^{N \times d}$, which is also normalized. Mathematically, we have:

$$\hat{y} = \arg\max_{c \in N} TW_c^T.\tag{2}$$

In our method for class removal, the projection matrix $P$ plays a crucial role in altering the textual representations to forget specific classes.

**LoRA** LoRA is a method that introduces trainable low-rank matrices into the model, allowing it to capture task-specific features without significantly modifying the original pre-trained model weights. This approach enables efficient fine-tuning by maintaining the majority of the model's learned knowledge while adapting to new tasks through a lightweight set of parameters. LoRA assumes that the weight updates for a pre-trained model can be constrained to a low-rank subspace. Given a weight matrix $P_0 \in \mathbb{R}^{k \times d}$ in the model with $k$ the input and $d$ the output dimensions, LoRA modifies it as $P = P_0 + \Delta P$ where the change in the original $P_0$ matrix, $\Delta P$, is decomposed into a low-rank form $\Delta P = AB^T$ with $A \in \mathbb{R}^{k \times r}$ and $B \in \mathbb{R}^{d \times r}$ being low-rank matrices with $r \ll min(d, k)$. Only $A$ and $B$ are optimized while the original $P_0$ is kept frozen.

## 4 Method

**Loss**  Our method relies on an optimization approach that directly considers an explicit low-rank adaptation of the text projection matrix into the shared image-text representation space. The main principles we use are that the image-text representation space should be minimally changed for the classes that are to be retained and should be changed for the classes that are to be forgotten in a systematic manner such that forgetting is achieved. We also want the low-rank transformation matrices to be sparse to increase efficiency. These requirements lead to direct terms in our optimization approach. Note that similar optimization approaches for forgetting have been used previously for instance in LLM (Li et al., 2024b) and Stable Diffusion concept editing (Gandikota et al., 2024). However, our primary contribution does not lie specifically in the optimization approach. Instead, our main contribution is a straightforward method for achieving class forgetting in CLIP by applying constraints to the text projection matrix. This enables us to achieve forgetting in CLIP without requiring the actual data used for training (which is not available for CLIP). Further, our approach differs in being a low-rank adaptation that provides an explicit forgetting to be achieved as the change in projection is known precisely through the low-rank projection adaptation. As we only change one matrix we can track what is being changed by our algorithm - we find that there exist some "magic" neurons that the method targets in a specific manner to forget. This analysis is provided in Section 6.4. Also, we show in the Appendix that there is a positive relation between the difficulty of forgetting a class and the Frobenious norm in the matrix of weights change.

Given CLIP textual encoder $f_\theta$ that encodes input text into its vector representation and $P$ the projection matrix that projects this representation into the image-text shared space, we optimize the low-rank update to $P$ given by the product of low-rank matrices $A$ and $B$. We aim to minimize the following loss:

$$
\begin{aligned}
\mathcal{L} = \ &\lambda_1 \left\| (P + \Delta P) f_\theta(X_r) - P f_\theta(X_r) \right\|_2^2 + \\
&\lambda_2 \left\| (P + \Delta P) f_\theta(X_f) - F \right\|_2^2 + \\
&\lambda_3 \left\| \Delta P - \mathbf{0} \right\|_2^2,
\end{aligned}
\tag{3}
$$

where $\Delta P = AB^\top$, $X_r$ represents the textual classes to retain, $X_f$ represents textual classes to forget and $F$ is the new representation of the forget class that we discuss below. The first component of the loss ensures that classes to retain are maintained close to their original position in the embedding space. The second component modifies the projection matrix $P$ such that the class to forget is projected into a new position of the image-text space $F$. The third component ensures that this is done with minimum modification to $P$. We cannot include all classes seen by CLIP during its contrastive pre-training in $X_r$ since these are unknown. However, we find that including semantically similar classes to the forget class suffices to keep the representations of all the retain textual classes we tested on fairly untouched. Indeed, it is important to include semantically similar classes because when forgetting a class we perturb the space around that class which affects representation of similar classes, thus preserving those ensures that only the forget class is projected to a different part of the image-text space while retaining classes that were close to it in the embedding space. As we show in the ablations in Section 7.3, retaining any type of classes is not useful as it reduces the performance on classes of the dataset the forget class was picked from. To generate semantically similar classes, we use a large language model (LLM) using a prompt *"Generate semantically similar classes to {class}"*. These are shown in the Appendix.

To determine where to project the forget class, denoted $F$ in Eq. 3, we use the empty token representation. In the ablations in Section 7.5 we test other variations such as a random projection and a perturbed representation of the forget class, which led to slightly worse results.

**Determination of the Loss Parameters**  We fix $\lambda_1$ and $\lambda_3$ while $\lambda_2$ is is determined iteratively. At each iteration, we assess the reduction in the second component of the loss to evaluate whether the change in the projection matrix $P$ is sufficient to project the forget class to the new chosen vector. We start from a fixed $\lambda_2$ and increment it in small steps until the reduction in the second loss component exceeds 0.75% of its initial value. Additional implementation details are described in the Appendix.

## 5 Experiments

### 5.1 Comparable Methods

There exist only one directly comparable method on CLIP class forgetting, while other are adapted from other methods. We only compare our approach to zero-shot methods that do not require any real data.

**Lipschitz CLIP Class Forgetting (Lip)** To forget specific classes (Kravets & Namboodiri, 2025) locally perturb both image and text representation of the forget class by a Gaussian noise and minimize the Lipschitz regularization loss updating both the encoders. The method is zero-shot because, instead of the original images, synthetic images generated by gradient ascent are utilized.

**Embedding regularization loss (Emb)** Similar to the above, instead of Lipschitz regularization loss a simple difference between embeddings with L2 regularization term is used.

**Amnesiac forgetting with synthetic data (Amns)** The approach from Graves et al. (2021) is adapted to a multi-modal setting by fine-tuning CLIP using the same contrastive loss employed in its initial training. In this approach, the labels of the classes to forget are randomly replaced with different labels using synthetic data. To maintain zero-shot setting, data from the classes to retain are not utilized and solely data for the class to forget are employed to forget.

**Error Minimization-Maximization Noise (EMMN)** The approach from Chundawat et al. (2023) is adapted to multi-modal setting learning retain and forget samples through loss minimization and maximization respectively and training the model on these samples.

### 5.2 Datasets

Following (Kravets & Namboodiri, 2025) we evaluate CLIP's forgetting capabilities on four high-quality, fine-grained datasets: Caltech101 (Fei-Fei et al., 2007) contains images from 101 distinct categories, each representing various objects or scenes. StanfordCars (Krause et al., 2013) contains images of cars of different makes and models. OxfordFlowers (Nilsback & Zisserman, 2008) includes images of flowers of 102 different classes. StanfordDogs (Khosla et al., 2011) comprises 120 classes of dogs of different species.

### 5.3 Evaluation

Ideally, to assess the forgetting procedure we should compare against the retrained model without the forget class. However, as CLIP training data are unknown and even if they were open sourced the computational power required to assess against a retrained model would be prohibitive, we adopt a similar logic to (Kravets & Namboodiri, 2025) in order to assess how well the class has been forgotten. We want the accuracy on the forget class to be as low as possible while maintaining the accuracy on other classes to a similar level before forgetting. As we need to compare different quantities such as the drop in accuracy of the forget class, the remaining accuracy of the dataset the class was picked from and remaining accuracy on other datasets we create an aggregated metrics for an easier comparison. Given the normalized reduction in the accuracy of the class to forget $A_{cl}$ and normalized reduction in the accuracy on the remaining classes for the $N$ examined datasets, each denoted as $A_{\{ds\}}$, where one of those $N$ datasets is the dataset the forget class was picked from, we calculate the **Average Score** Kravets & Namboodiri (2025) metrics as:

$$\text{Avg. Score} = \frac{1}{N+1}((1 - A_{cl}) + \sum_{ds} A_{\{ds\}}). \tag{4}$$

Best methods will have a **small** average score. During evaluation we use the standard template *A photo of a {class}*, however in the ablations we evaluate the forget model with other templates to test the robustness to different evaluation templates.

It is important to note that forgetting is a broad concept that cannot be easily guaranteed and there are multiple metrics available to evaluate it. We use forget accuracy (for classification) and precision@K (for

Table 1: Main forgetting results. We compare our method to four other methods averaging across three classes for four selected datasets.

| Method | Model | Dataset | Avg. Target Class acc. | | Avg. Other Classes acc. | | Avg. StanfordCars | | Avg. StanfordDogs | | Avg. Caltech101 | | Avg. OxfordFlowers | | Avg. Score (↓) |
|---|---|---|---|---|---|---|---|---|---|---|---|---|---|---|---|
| | | | BF | AF | BF | AF | BF | AF | BF | AF | BF | AF | BF | AF | |
| Ours | RN50 | StanfordCars | 0.397 | 0.0 | 0.558 | 0.55 | - | - | 0.517 | 0.51 | 0.857 | 0.855 | 0.661 | 0.657 | **0.007** |
| Lip | RN50 | StanfordCars | 0.397 | 0.056 | 0.558 | 0.551 | - | - | 0.517 | 0.513 | 0.857 | 0.86 | 0.661 | 0.653 | 0.034 |
| Emb | RN50 | StanfordCars | 0.397 | 0.087 | 0.558 | 0.536 | - | - | 0.517 | 0.51 | 0.857 | 0.85 | 0.661 | 0.649 | 0.06 |
| Amns | RN50 | StanfordCars | 0.397 | 0.357 | 0.558 | 0.498 | - | - | 0.517 | 0.505 | 0.857 | 0.863 | 0.661 | 0.653 | 0.208 |
| EMMN | RN50 | StanfordCars | 0.397 | 0.0 | 0.558 | 0.054 | - | - | 0.517 | 0.043 | 0.857 | 0.424 | 0.661 | 0.069 | 0.644 |
| Ours | RN50 | StanfordDogs | 0.593 | 0.0 | 0.516 | 0.509 | 0.558 | 0.554 | - | - | 0.857 | 0.856 | 0.661 | 0.653 | **0.007** |
| Lip | RN50 | StanfordDogs | 0.593 | 0.048 | 0.516 | 0.516 | 0.558 | 0.558 | - | - | 0.857 | 0.866 | 0.661 | 0.655 | 0.018 |
| Emb | RN50 | StanfordDogs | 0.593 | 0.261 | 0.516 | 0.479 | 0.558 | 0.554 | - | - | 0.857 | 0.836 | 0.661 | 0.621 | 0.121 |
| Amns | RN50 | StanfordDogs | 0.593 | 0.327 | 0.516 | 0.465 | 0.558 | 0.556 | - | - | 0.857 | 0.848 | 0.661 | 0.643 | 0.138 |
| EMMN | RN50 | StanfordDogs | 0.593 | 0.0 | 0.516 | 0.053 | 0.558 | 0.107 | - | - | 0.857 | 0.493 | 0.661 | 0.107 | 0.594 |
| Ours | RN50 | Caltech101 | 0.839 | 0.0 | 0.857 | 0.859 | 0.558 | 0.56 | 0.517 | 0.513 | - | - | 0.661 | 0.658 | **0.002** |
| Lip | RN50 | Caltech101 | 0.839 | 0.081 | 0.857 | 0.865 | 0.558 | 0.557 | 0.517 | 0.52 | - | - | 0.661 | 0.657 | 0.021 |
| Emb | RN50 | Caltech101 | 0.839 | 0.131 | 0.857 | 0.83 | 0.558 | 0.546 | 0.517 | 0.501 | - | - | 0.661 | 0.618 | 0.061 |
| Amns | RN50 | Caltech101 | 0.838 | 0.33 | 0.857 | 0.834 | 0.558 | 0.553 | 0.517 | 0.502 | - | - | 0.661 | 0.627 | 0.102 |
| EMMN | RN50 | Caltech101 | 0.839 | 0.0 | 0.857 | 0.397 | 0.558 | 0.097 | 0.517 | 0.081 | - | - | 0.661 | 0.13 | 0.602 |
| Ours | RN50 | OxfordFlowers | 0.848 | 0.0 | 0.659 | 0.651 | 0.558 | 0.558 | 0.517 | 0.515 | 0.857 | 0.858 | - | - | **0.003** |
| Lip | RN50 | OxfordFlowers | 0.848 | 0.0 | 0.659 | 0.645 | 0.558 | 0.557 | 0.517 | 0.509 | 0.857 | 0.868 | - | - | 0.008 |
| Emb | RN50 | OxfordFlowers | 0.848 | 0.442 | 0.659 | 0.625 | 0.558 | 0.553 | 0.517 | 0.5 | 0.857 | 0.85 | - | - | 0.122 |
| Amns | RN50 | OxfordFlowers | 0.848 | 0.388 | 0.659 | 0.592 | 0.558 | 0.54 | 0.517 | 0.487 | 0.857 | 0.835 | - | - | 0.135 |
| EMMN | RN50 | OxfordFlowers | 0.848 | 0.0 | 0.659 | 0.121 | 0.558 | 0.121 | 0.517 | 0.112 | 0.857 | 0.676 | - | - | 0.519 |
| Ours | ViT-B/16 | StanfordCars | 0.595 | 0.0 | 0.656 | 0.642 | - | - | 0.591 | 0.591 | 0.933 | 0.934 | 0.708 | 0.703 | **0.006** |
| Lip | ViT-B/16 | StanfordCars | 0.595 | 0.159 | 0.656 | 0.642 | - | - | 0.591 | 0.584 | 0.933 | 0.932 | 0.708 | 0.707 | 0.06 |
| Emb | ViT-B/16 | StanfordCars | 0.595 | 0.0 | 0.656 | 0.557 | - | - | 0.591 | 0.508 | 0.933 | 0.921 | 0.708 | 0.69 | 0.066 |
| Amns | ViT-B/16 | StanfordCars | 0.595 | 0.143 | 0.656 | 0.18 | - | - | 0.591 | 0.398 | 0.933 | 0.876 | 0.708 | 0.51 | 0.327 |
| EMMN | ViT-B/16 | StanfordCars | 0.595 | 0.159 | 0.656 | 0.182 | - | - | 0.591 | 0.119 | 0.933 | 0.589 | 0.708 | 0.137 | 0.592 |
| Ours | ViT-B/16 | StanfordDogs | 0.673 | 0.0 | 0.591 | 0.582 | 0.655 | 0.653 | - | - | 0.933 | 0.93 | 0.708 | 0.697 | **0.008** |
| Lip | ViT-B/16 | StanfordDogs | 0.673 | 0.142 | 0.591 | 0.592 | 0.655 | 0.647 | - | - | 0.933 | 0.935 | 0.708 | 0.709 | 0.045 |
| Emb | ViT-B/16 | StanfordDogs | 0.673 | 0.071 | 0.591 | 0.518 | 0.655 | 0.632 | - | - | 0.933 | 0.93 | 0.708 | 0.699 | 0.056 |
| Amns | ViT-B/16 | StanfordDogs | 0.673 | 0.219 | 0.591 | 0.358 | 0.655 | 0.59 | - | - | 0.933 | 0.901 | 0.708 | 0.572 | 0.209 |
| EMMN | ViT-B/16 | StanfordDogs | 0.673 | 0.042 | 0.591 | 0.365 | 0.655 | 0.284 | - | - | 0.933 | 0.826 | 0.708 | 0.438 | 0.301 |
| Ours | ViT-B/16 | Caltech101 | 0.971 | 0.0 | 0.933 | 0.932 | 0.655 | 0.653 | 0.591 | 0.574 | - | - | 0.708 | 0.699 | **0.009** |
| Lip | ViT-B/16 | Caltech101 | 0.971 | 0.576 | 0.933 | 0.935 | 0.655 | 0.652 | 0.591 | 0.594 | - | - | 0.708 | 0.709 | 0.12 |
| Emb | ViT-B/16 | Caltech101 | 0.971 | 0.598 | 0.933 | 0.91 | 0.655 | 0.609 | 0.591 | 0.517 | - | - | 0.708 | 0.656 | 0.182 |
| Amns | ViT-B/16 | Caltech101 | 0.971 | 0.846 | 0.933 | 0.848 | 0.655 | 0.517 | 0.591 | 0.445 | - | - | 0.708 | 0.533 | 0.334 |
| EMMN | ViT-B/16 | Caltech101 | 0.971 | 0.284 | 0.933 | 0.813 | 0.655 | 0.352 | 0.591 | 0.302 | - | - | 0.708 | 0.473 | 0.341 |
| Ours | ViT-B/16 | OxfordFlowers | 0.784 | 0.0 | 0.707 | 0.705 | 0.655 | 0.654 | 0.591 | 0.584 | 0.933 | 0.933 | - | - | **0.004** |
| Lip | ViT-B/16 | OxfordFlowers | 0.784 | 0.078 | 0.707 | 0.702 | 0.655 | 0.645 | 0.591 | 0.588 | 0.933 | 0.933 | - | - | 0.026 |
| Emb | ViT-B/16 | OxfordFlowers | 0.784 | 0.0 | 0.707 | 0.617 | 0.655 | 0.543 | 0.591 | 0.522 | 0.933 | 0.906 | - | - | 0.089 |
| Amns | ViT-B/16 | OxfordFlowers | 0.784 | 0.834 | 0.707 | 0.527 | 0.655 | 0.602 | 0.591 | 0.526 | 0.933 | 0.913 | - | - | 0.307 |
| EMMN | ViT-B/16 | OxfordFlowers | 0.784 | 0.02 | 0.707 | 0.433 | 0.655 | 0.317 | 0.591 | 0.304 | 0.933 | 0.83 | - | - | 0.305 |

retrieval) to measure the effectiveness of forgetting. However, other metrics such as Membership Inference Attacks (MIA) (Shokri et al., 2017) exist and offer different perspectives. In our case, we cannot apply MIA because it is typically used to check if specific training data remains embedded in the model. Since we do not have access to the training data used in CLIP's pre-training, applying MIA is not feasible in this context.

# 6 Results

## 6.1 Comparison against other forgetting methods

In Tab.1 we present the aggregated forgetting results across different methods with RN50 and ViT-B/16 visual encoders respectively. To compute them we first forget each of the 3 selected classes (can be found in Tab. 2) individually from each dataset and then average the results over these 3 classes. Granular results are found in the Appendix. *Method* column indicates the forgetting method used, *Dataset* column indicates the dataset from which the class to be forgotten was picked. The *Avg. Target Class acc.* column denotes the accuracy on the target class averaged among the 3 selected classes when forgetting each class individually while *Avg. Other Classes acc.* indicates the average accuracy on the remaining classes, excluding the selected forgotten class, also averaged while forgetting each class individually from the corresponding dataset. These results are shown before (BF) and after (AF) forgetting. The last eight columns represent the results on the remaining datasets reported both before and after forgetting. It's worth noting that *Avg. Other Classes acc.* measures the model's accuracy on the remaining classes from *Dataset* that are **semantically similar**

Table 2: Forgetting on multiple classes with RN50 and ViT-B/16 models.

| Method | Model | Dataset | Classes | Avg. Target Classes acc. | | Other Classes acc. | | StanfordCars | | StanfordDogs | | Caltech101 | | OxfordFlowers | | Avg. Score (↓) |
|---|---|---|---|---|---|---|---|---|---|---|---|---|---|---|---|---|
| | | | | BF | AF | BF | AF | BF | AF | BF | AF | BF | AF | BF | AF | |
| Lip | RN50 | StanfordDogs | Pekinese,toy poodle,Scotch terrier | 0.591 | 0.091 | 0.515 | 0.507 | 0.558 | 0.547 | - | - | 0.857 | 0.865 | 0.661 | 0.633 | 0.046 |
| Lip | RN50 | StanfordCars | 2009 Spyker C8 Coupe, 2010 Dodge Ram Pickup 3500 Crew Cab, 2011 Ford Ranger SuperCab | 0.397 | 0.222 | 0.56 | 0.519 | - | - | 0.517 | 0.482 | 0.857 | 0.84 | 0.661 | 0.607 | 0.16 |
| Lip | RN50 | Caltech101 | euphonium,minaret,platypus | 0.827 | 0.125 | 0.858 | 0.869 | 0.558 | 0.549 | 0.517 | 0.515 | - | - | 0.661 | 0.633 | 0.042 |
| Lip | RN50 | OxfordFlowers | gazania,tree mallow,trumpet creeper | 0.86 | 0.0 | 0.656 | 0.609 | 0.558 | 0.552 | 0.517 | 0.498 | 0.857 | 0.863 | - | - | 0.023 |
| Ours | RN50 | StanfordDogs | Pekinese,toy poodle,Scotch terrier | 0.591 | 0.0 | 0.515 | 0.499 | 0.558 | 0.54 | - | - | 0.857 | 0.854 | 0.661 | 0.629 | **0.023** |
| Ours | RN50 | StanfordCars | 2009 Spyker C8 Coupe, 2010 Dodge Ram Pickup 3500 Crew Cab, 2011 Ford Ranger SuperCab | 0.397 | 0.0 | 0.56 | 0.53 | - | - | 0.517 | 0.499 | 0.857 | 0.85 | 0.661 | 0.654 | **0.021** |
| Ours | RN50 | Caltech101 | euphonium,minaret,platypus | 0.827 | 0.0 | 0.858 | 0.863 | 0.558 | 0.551 | 0.517 | 0.499 | - | - | 0.661 | 0.655 | **0.011** |
| Ours | RN50 | OxfordFlowers | trumpet creeper,gazania,tree mallow | 0.86 | 0.0 | 0.656 | 0.627 | 0.558 | 0.554 | 0.517 | 0.502 | 0.857 | 0.856 | - | - | **0.016** |
| Lip | ViT-B/16 | StanfordDogs | Pekinese,toy poodle,Scotch terrier | 0.672 | 0.251 | 0.589 | 0.584 | 0.655 | 0.644 | - | - | 0.933 | 0.939 | 0.708 | 0.713 | 0.08 |
| Lip | ViT-B/16 | StanfordCars | 2009 Spyker C8 Coupe, 2010 Dodge Ram Pickup 3500 Crew Cab, 2011 Ford Ranger SuperCab | 0.595 | 0.3 | 0.656 | 0.625 | - | - | 0.591 | 0.576 | 0.933 | 0.928 | 0.708 | 0.699 | 0.119 |
| Lip | ViT-B/16 | Caltech101 | euphonium,minaret,platypus | 0.971 | 0.498 | 0.932 | 0.929 | 0.655 | 0.634 | 0.591 | 0.589 | - | - | 0.708 | 0.709 | 0.11 |
| Lip | ViT-B/16 | OxfordFlowers | trumpet creeper,gazania,tree mallow | 0.807 | 0.31 | 0.705 | 0.68 | 0.655 | 0.613 | 0.591 | 0.551 | 0.933 | 0.929 | - | - | 0.111 |
| Ours | ViT-B/16 | StanfordDogs | Pekinese,toy poodle,Scotch terrier | 0.672 | 0.0 | 0.589 | 0.557 | 0.655 | 0.624 | - | - | 0.933 | 0.92 | 0.708 | 0.668 | **0.035** |
| Ours | ViT-B/16 | StanfordCars | 2009 Spyker C8 Coupe, 2010 Dodge Ram Pickup 3500 Crew Cab, 2011 Ford Ranger SuperCab | 0.595 | 0.0 | 0.656 | 0.633 | - | - | 0.591 | 0.586 | 0.933 | 0.931 | 0.708 | 0.693 | **0.013** |
| Ours | ViT-B/16 | Caltech101 | euphonium,minaret,platypus | 0.962 | 0.0 | 0.932 | 0.929 | 0.655 | 0.65 | 0.591 | 0.558 | - | - | 0.708 | 0.685 | **0.02** |
| Ours | ViT-B/16 | OxfordFlowers | trumpet creeper,gazania,tree mallow | 0.807 | 0.0 | 0.705 | 0.682 | 0.655 | 0.649 | 0.591 | 0.578 | 0.933 | 0.929 | - | - | **0.014** |

to the forget class (e.g. forgetting "Pekinese" and testing on other breeds of dogs) while the final columns measure model's accuracy on **semantically different** classes. It's important to test both because, even if normally neighboring concepts of the forget class are the most affected by the forgetting procedure, sometimes semantically different classes might be affected as well. We show additional experiments in multi-label setting and with more diverse class distribution in the Appendix.

For both models, we observe that our method achieves superior performance in terms of the average score, balancing effective forgetting of the target class while retaining information about the not targeted classes. Specifically, our method is able to remove completely the information about the forget classes from the model while other methods, apart from EMNN that however overforgets other classes, usually retain some information. *Lip* is the most competitive with our method that sometimes achieves better accuracy on the other classes retaining however some information about the forget class. Furthermore, comparing to *Lip*, which often struggles to forget well with a ViT visual encoder, our method is more robust and forgets well independently on the visual encoder used. Full results can be found in the Appendix. In the Appendix we also evaluate retrieval results and make some additional considerations.

## 6.2 Identity Forgetting

We validate our methodology on identity forgetting in line with the motivation of the right to be forgotten. We use PinsFaces (Burak) dataset that contains 105 celebrity faces for this purpose. These results are presented in the Tab. 3 where we observe that our method works to forget different identities as well.

Table 3: Forgetting identities from PinsFaces dataset.

| Model | Class name | Target Class acc. | | Other Classes acc. | | StanfordCars | | StanfordDogs | | Caltech101 | | OxfordFlowers | | Avg. Score (↓) |
|---|---|---|---|---|---|---|---|---|---|---|---|---|---|---|
| | | BF | AF | BF | AF | BF | AF | BF | AF | BF | AF | BF | AF | |
| RN50 | Henry Cavil | 0.866 | 0.0 | 0.82 | 0.804 | 0.558 | 0.558 | 0.517 | 0.512 | 0.857 | 0.861 | 0.661 | 0.661 | 0.006 |
| RN50 | Amanda Crew | 0.828 | 0.0 | 0.821 | 0.813 | 0.558 | 0.559 | 0.517 | 0.514 | 0.857 | 0.858 | 0.661 | 0.654 | 0.005 |
| RN50 | Gal Gadot | 0.707 | 0.0 | 0.822 | 0.815 | 0.558 | 0.562 | 0.517 | 0.516 | 0.857 | 0.853 | 0.661 | 0.659 | 0.003 |
| ViT-B/16 | Henry Cavil | 0.959 | 0.0 | 0.907 | 0.902 | 0.655 | 0.647 | 0.591 | 0.574 | 0.933 | 0.933 | 0.708 | 0.707 | 0.01 |
| ViT-B/16 | Amanda Crew | 1.0 | 0.0 | 0.907 | 0.903 | 0.655 | 0.653 | 0.591 | 0.587 | 0.933 | 0.933 | 0.708 | 0.702 | 0.005 |
| ViT-B/16 | Gal Gadot | 0.879 | 0.0 | 0.908 | 0.902 | 0.655 | 0.659 | 0.591 | 0.589 | 0.933 | 0.932 | 0.708 | 0.709 | 0.002 |

## 6.3 Forgetting on Multiple Classes

In Tab. 2 we show the results for *Lip* and *Our* methods when performing forgetting on multiple classes for RN50 and ViT-B/16 visual encoders respectively. Our method shows its superiority in terms of the average

score also in this case. Again, our method is able to completely forget all the targeted classes while still maintaining high accuracy across not targeted classes while *Lip*, especially with the ViT-B/16 visual encoder retains substantial information on the classes to be forgotten. Indeed, *Lip* is less consistent across different architectures while our method is able to maintain this consistency in both single-class and multiple-class forgetting. We also show in the Appendix we that even if the forgetting requests for multiple classes arrive sequentially, our method performs well.

## 6.4 Understanding the Forgetting

Thanks to the simplicity of our method that only modifies one matrix we can closely examine what happens during the forgetting process. Specifically, we analyze which neurons in the projection matrix undergo the most significant changes following the forgetting procedure. This is done by looking at the absolute value of the $AB^T$ matrix that represents the changes in the text projection matrix. Recall that all the projection matrix does is projecting the hidden textual representation into the text-image shared embedding space: from 512 into 1024 dimensions for CLIP with ResNet50 visual encoder and from 512 into 512 dimensions with ViT-B/16. We observe that there are some "magic" neurons that the algorithm modifies more indicating that these textual features need to change the most to forget a class while preserving the other classes. For example, for ResNet50 such neurons are in column 222 of the the weight projection change matrix $(AB^T)_{[512,1024]}$ as can be seen in Fig. 2 that shows on the x axis the column where most change occurred and on the y axis the sum of absolute values of the changes in that column.

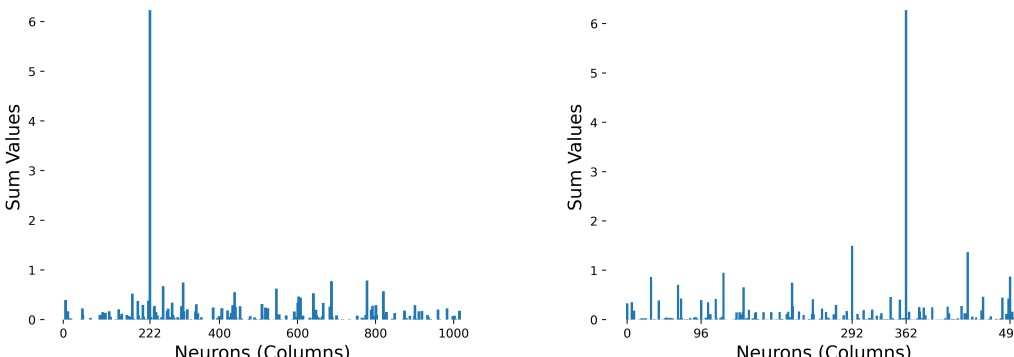

Figure 2: Sum of absolute values of neurons in different columns of the textual projection matrix change. On the **left** for RN50 visual encoder, on the **right** for ViT-B/16 visual encoder.

It turns out this is not a random selection; plotting textual features across different classes and datasets, shown in Fig. 3, reveals that feature 222, which the forgetting algorithm targets, has the largest value. Thus, changing this feature is the easiest way for the model to forget a class, decreasing the dot product between the visual and textual features for that class. The corresponding visual feature 222 is also negative across images, so the network increases the value of the textual feature 222, decreasing the dot product for that class causing a change in model's prediction. The presence of such neurons and the algorithm's targeted modification is intriguing. A similar phenomenon occurs with the ViT-B/16 as seen in the same figure. We provide a more detailed example in the Appendix.

We do a similar analysis for Lipschitz forgetting, where tracking changes at different weight levels becomes challenging due to the modification of many layers. However, we can observe the alterations in the final visual and textual features. Interestingly, when examining the features that change the most with Lipschitz forgetting, we observe the same pattern as with our method for the textual features. In contrast, this behavior is not seen for the visual features where different features undergo more significant changes for different images.

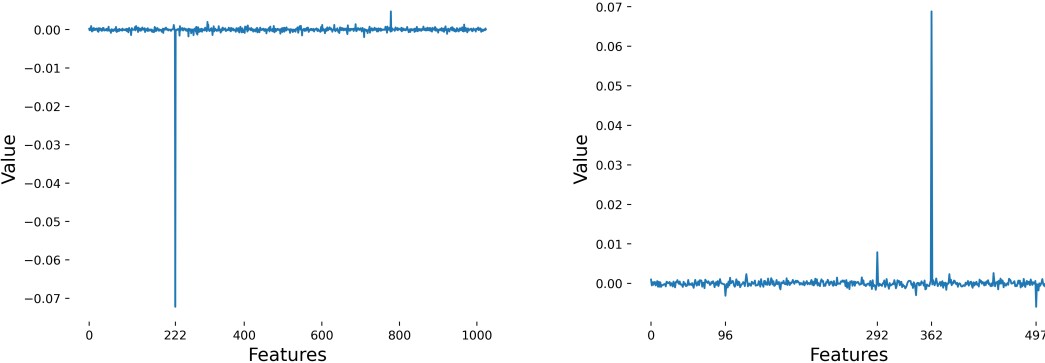

Figure 3: Textual features values averaged across different classes. On the **left** for RN50 visual encoder, on the **right** for ViT-B/16 visual encoder.

## 6.5 Forgetting is Specific

The datasets in our main experiments include semantically similar classes that often share words. Our procedure effectively breaks the exact textual and visual association for the forget class but doesn't break the association for similar words to the forget class as evidenced by the high accuracy on other classes from the same dataset that are often similar in meaning to the forget class. For instance, when removing the class information for *toy poodle*, classes *miniature poodle* and *standard poodle*, which share the word *poodle*, maintain accuracy comparable to that before forgetting. To eliminate synonyms and similar words, the forgetting procedure would need to be repeated for those terms. We see this as a feature rather than a limitation, as it enables the preservation of as much information as possible during the forgetting process. This approach provides precise control over which information to forget, including synonyms and similar words if necessary.

In Fig. 4 we show examples of the model's predictions before (BF) and after forgetting (AF). We observe that the new classes predicted by the model after forgetting are close to the correct ones indicating that our method targets specific knowledge of the model while preserving its general understanding.

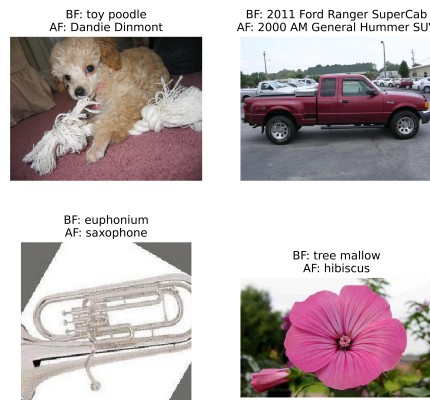

Figure 4: Predictions before (BF) and after forgetting (AF) with the prediction BF representing the target class to forget.

## 7 Ablations

### 7.1 Variation of Templates for Evaluation

In these experiments we test how sensitive the model after forgetting is to the evaluation template and whether when changing it the model is still able to retrieve the forget class. Following (Kravets & Namboodiri, 2025) we evaluate using the following three templates: *"We can see a {class} in this image"*, *"This is a representation of {class}"*, *"There is evidence of a {class} in the picture"*. Note that by changing the evaluation template also the accuracy of zero-shot CLIP changes. In Tab. 4 we observe that forgetting is robust to the change in the evaluation template as the model is still unable to retrieve the forget class and maintains a high accuracy of the not forget classes relatively to the model before forgetting.

Table 4: Aggregated results across different evaluation templates. We aggregate across 3 evaluations template to assess sensitivity of the models after forgetting to the change in the evaluation template.

| Model | Dataset | Avg. Target Class acc. | | Avg. Other Classes acc. | | Avg. StanfordCars | | Avg. StanfordDogs | | Avg. Caltech101 | | Avg. OxfordFlowers | | Avg. Score (↓) |
|---|---|---|---|---|---|---|---|---|---|---|---|---|---|---|
| | | BF | AF | BF | AF | BF | AF | BF | AF | BF | AF | BF | AF | |
| RN50 | StanfordCars | 0.272 | 0.0 | 0.493 | 0.488 | - | - | 0.415 | 0.415 | 0.81 | 0.811 | 0.519 | 0.518 | 0.003 |
| RN50 | StanfordDogs | 0.306 | 0.0 | 0.416 | 0.405 | 0.492 | 0.493 | - | - | 0.81 | 0.809 | 0.519 | 0.518 | 0.007 |
| RN50 | Caltech101 | 0.879 | 0.029 | 0.81 | 0.81 | 0.492 | 0.488 | 0.415 | 0.415 | - | - | 0.519 | 0.517 | 0.01 |
| RN50 | OxfordFlowers | 0.698 | 0.0 | 0.518 | 0.513 | 0.492 | 0.491 | 0.415 | 0.415 | 0.81 | 0.81 | - | - | 0.004 |
| ViT-B/16 | StanfordCars | 0.497 | 0.0 | 0.623 | 0.618 | - | - | 0.516 | 0.514 | 0.88 | 0.882 | 0.61 | 0.61 | 0.003 |
| ViT-B/16 | StanfordDogs | 0.532 | 0.0 | 0.516 | 0.504 | 0.622 | 0.617 | - | - | 0.88 | 0.88 | 0.61 | 0.607 | 0.008 |
| ViT-B/16 | Caltech101 | 0.97 | 0.011 | 0.879 | 0.88 | 0.622 | 0.621 | 0.516 | 0.513 | - | - | 0.61 | 0.61 | 0.004 |
| ViT-B/16 | OxfordFlowers | 0.667 | 0.0 | 0.609 | 0.604 | 0.622 | 0.62 | 0.516 | 0.513 | 0.88 | 0.88 | - | - | 0.004 |

## 7.2 Loss Components Ablation

In this subsection we assess how important are the loss components to the forgetting procedure. For this, we first set $\lambda_1$ to 0 and then $\lambda_3$ to 0. Results are shown in Tab. 5 where we observe that both components are important for forgetting that achieve the best results in terms of the average score when all the components are included. The average score drops more when $\lambda_1$ is excluded from the loss.

Table 5: Ablations on the importance of the different components of the loss setting $\lambda_1$ to 0 or $\lambda_3$ to 0.

| Method | Model | Avg. Target Class acc. | | Avg. Other Classes acc. | | Avg. StanfordCars | | Avg. StanfordDogs | | Avg. Caltech101 | | Avg. OxfordFlowers | | Avg. Score (↓) |
|---|---|---|---|---|---|---|---|---|---|---|---|---|---|---|
| | | BF | AF | BF | AF | BF | AF | BF | AF | BF | AF | BF | AF | |
| All loss terms | RN50 | 0.669 | 0.0 | 0.648 | 0.642 | 0.558 | 0.557 | 0.517 | 0.513 | 0.857 | 0.857 | 0.661 | 0.656 | **0.005** |
| Excluding $\lambda_1$ | RN50 | 0.669 | 0.01 | 0.648 | 0.625 | 0.558 | 0.558 | 0.517 | 0.511 | 0.857 | 0.863 | 0.661 | 0.644 | 0.017 |
| Excluding $\lambda_3$ | RN50 | 0.669 | 0.004 | 0.648 | 0.641 | 0.558 | 0.553 | 0.517 | 0.509 | 0.857 | 0.855 | 0.661 | 0.655 | 0.01 |
| All loss terms | ViT-B/16 | 0.756 | 0.0 | 0.722 | 0.715 | 0.655 | 0.653 | 0.591 | 0.583 | 0.933 | 0.932 | 0.708 | 0.7 | **0.008** |
| Excluding $\lambda_1$ | ViT-B/16 | 0.756 | 0.185 | 0.722 | 0.694 | 0.655 | 0.653 | 0.591 | 0.587 | 0.933 | 0.935 | 0.708 | 0.702 | 0.061 |
| Excluding $\lambda_3$ | ViT-B/16 | 0.756 | 0.001 | 0.722 | 0.711 | 0.655 | 0.636 | 0.591 | 0.578 | 0.933 | 0.928 | 0.708 | 0.694 | 0.019 |

## 7.3 Number of Classes to Retain

In this section we evaluate how varying the number of classes to preserve affects the forgetting results. In Tab. 6 we observe that reducing the number of classes to preserve the average score drops, but the results are still relatively robust even when only 10% classes to retain are used - the most sensitive to the reduction in retain classes is the dataset the forget class was picked from. All the generated retain classes can be found in the Appendix, and on average we generate 100 semantically similar classes to the forget class for each dataset.

Table 6: Ablations on the number of classes to retain.

| % Number of Classes | Model | Avg. Target Class acc. | | Avg. Other Classes acc. | | Avg. StanfordCars | | Avg. StanfordDogs | | Avg. Caltech101 | | Avg. OxfordFlowers | | Avg. Score (↓) |
|---|---|---|---|---|---|---|---|---|---|---|---|---|---|---|
| | | BF | AF | BF | AF | BF | AF | BF | AF | BF | AF | BF | AF | |
| All | RN50 | 0.669 | 0.0 | 0.648 | 0.642 | 0.558 | 0.556 | 0.517 | 0.509 | 0.857 | 0.856 | 0.661 | 0.655 | **0.007** |
| 50% | RN50 | 0.669 | 0.0 | 0.648 | 0.636 | 0.558 | 0.554 | 0.517 | 0.51 | 0.857 | 0.856 | 0.661 | 0.653 | 0.01 |
| 10% | RN50 | 0.669 | 0.0 | 0.648 | 0.623 | 0.558 | 0.558 | 0.517 | 0.51 | 0.857 | 0.856 | 0.661 | 0.653 | 0.013 |
| All | ViT-B/16 | 0.756 | 0.0 | 0.722 | 0.712 | 0.655 | 0.646 | 0.591 | 0.576 | 0.933 | 0.93 | 0.708 | 0.695 | **0.015** |
| 50% | ViT-B/16 | 0.756 | 0.0 | 0.722 | 0.704 | 0.655 | 0.65 | 0.591 | 0.576 | 0.933 | 0.931 | 0.708 | 0.698 | 0.015 |
| 10% | ViT-B/16 | 0.756 | 0.0 | 0.722 | 0.688 | 0.655 | 0.652 | 0.591 | 0.581 | 0.933 | 0.932 | 0.708 | 0.7 | 0.016 |

## 7.4 Retaining Classes from Forget Class Dataset

In our main experiments we used semantically similar classes (*SemSim*) to the forget class for the retain loss component. In Tab. 7 we compare the effects of using actual classes from the dataset the forget class was picked from, denoted as $Cls_r$, and when using semantically different classes (*SemDiff*). Overall, we find

that semantically similar classes are crucial for maintaining high *Other Classes acc.*. When forgetting, the original projection matrix is altered in a way that perturbs the space near the forget class more leading to a greater reduction in accuracy for semantically similar classes which are closer in the image-text embedding space compared to different classes, where the space is less affected. Using actual classes ($Cls_r$) performs the best, but similarly to semantically similar classes generated by a large language model. In contrast, using semantically different classes, taken from the Food101 (Bossard et al., 2014) dataset, results in the worst outcome, especially for *Other Classes acc.* while the accuracy of the classes not semantically similar to the forget class (i.e. other test datasets) is maintained without explicitly including them.

Table 7: Ablations with actual ($Cls_r$), semantically similar ($SemSim$) and different ($SemDiff$) classes

| Type of Retained Classes | Model | Avg. Target Class acc. | | Avg. Other Classes acc. | | Avg. StanfordCars | | Avg. StanfordDogs | | Avg. Caltech101 | | Avg. OxfordFlowers | | Avg. Score ($\downarrow$) |
|---|---|---|---|---|---|---|---|---|---|---|---|---|---|---|
| | | BF | AF | BF | AF | BF | AF | BF | AF | BF | AF | BF | AF | |
| $Cls_r$ | RN50 | 0.669 | 0.0 | 0.648 | 0.644 | 0.558 | 0.558 | 0.517 | 0.51 | 0.857 | 0.856 | 0.661 | 0.659 | **0.004** |
| $SemSim$ | RN50 | 0.669 | 0.0 | 0.648 | 0.642 | 0.558 | 0.557 | 0.517 | 0.513 | 0.857 | 0.857 | 0.661 | 0.656 | 0.005 |
| $SemDiff$ | RN50 | 0.669 | 0.0 | 0.648 | 0.61 | 0.558 | 0.558 | 0.517 | 0.511 | 0.857 | 0.86 | 0.661 | 0.655 | 0.016 |
| $Cls_r$ | ViT-B/16 | 0.756 | 0.0 | 0.722 | 0.718 | 0.655 | 0.652 | 0.591 | 0.583 | 0.933 | 0.931 | 0.708 | 0.702 | **0.006** |
| $SemSim$ | ViT-B/16 | 0.756 | 0.0 | 0.722 | 0.715 | 0.655 | 0.653 | 0.591 | 0.583 | 0.933 | 0.932 | 0.708 | 0.7 | 0.008 |
| $SemDiff$ | ViT-B/16 | 0.756 | 0.004 | 0.722 | 0.68 | 0.655 | 0.651 | 0.591 | 0.584 | 0.933 | 0.934 | 0.708 | 0.703 | 0.018 |

## 7.5 Forget Class Projection

We evaluate the importance of where to project the forget classes in the shared image-text space. We test different variations like projecting into a random vector and a perturbed embedding of the forget concepts comparing them to the empty token projection used in our main experiments. In Tab 8 we see that projection space is less important as similar results are achieved when we project into different parts of the space.

Table 8: Ablations on projection. We perform an ablation study projecting into the empty token vector (*EmptyToken proj*), random vector sampled from Gaussian distribution (*Random proj*) and perturbed embedding of the forget class (*Perturbed proj*)

| Method | Model | Avg. Target Class acc. | | Avg. Other Classes acc. | | Avg. StanfordCars | | Avg. StanfordDogs | | Avg. Caltech101 | | Avg. OxfordFlowers | | Avg. Score ($\downarrow$) |
|---|---|---|---|---|---|---|---|---|---|---|---|---|---|---|
| | | BF | AF | BF | AF | BF | AF | BF | AF | BF | AF | BF | AF | |
| EmptyToken proj | RN50 | 0.669 | 0.0 | 0.648 | 0.642 | 0.558 | 0.557 | 0.517 | 0.513 | 0.857 | 0.857 | 0.661 | 0.656 | **0.005** |
| Random proj | RN50 | 0.669 | 0.0 | 0.648 | 0.642 | 0.558 | 0.557 | 0.517 | 0.511 | 0.857 | 0.856 | 0.661 | 0.655 | 0.006 |
| Perturbed proj | RN50 | 0.669 | 0.0 | 0.648 | 0.642 | 0.558 | 0.557 | 0.517 | 0.511 | 0.857 | 0.856 | 0.661 | 0.655 | 0.006 |
| EmptyToken proj | ViT-B/16 | 0.756 | 0.0 | 0.722 | 0.715 | 0.558 | 0.653 | 0.517 | 0.583 | 0.857 | 0.932 | 0.661 | 0.7 | **0.008** |
| Random proj | ViT-B/16 | 0.756 | 0.0 | 0.722 | 0.713 | 0.558 | 0.652 | 0.517 | 0.58 | 0.857 | 0.931 | 0.661 | 0.698 | 0.01 |
| Perturbed proj | ViT-B/16 | 0.756 | 0.0 | 0.722 | 0.713 | 0.558 | 0.652 | 0.517 | 0.58 | 0.857 | 0.931 | 0.661 | 0.698 | 0.01 |

## 8 Conclusions

In this work we demonstrated that it is possible to forget a class in the CLIP model without altering the original visual encoder, thereby eliminating the need to generate synthetic data. A learned adaptation to the projection matrix of the textual encoder, which projects textual representations into the image-text embedding space, is sufficient for class forgetting. Furthermore, we show that the representation of semantically similar classes can be affected during forgetting, reducing their accuracy. Therefore, it is crucial to include semantically similar classes that are close in the embedding space in the loss function to retain their information.

**Acknowledgements** We'd like to gratefully acknowledge Microsoft's compute support through Microsoft's Accelerating Foundation Models Research grant and the support from University of Bath for the studentship.

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

# Appendix

## Table of Contents

# A ResNet Full Results

Table 9: Forgetting results with RN50 visual encoder. We compare our methods with four others on three classes for four selected datasets.

| Method | Dataset | Class name | Target Class acc. | | Other Classes acc. | | StanfordCars | | StanfordDogs | | Caltech101 | | OxfordFlowers | |
|---|---|---|---|---|---|---|---|---|---|---|---|---|---|---|
| | | | BF | AF | BF | AF | BF | AF | BF | AF | BF | AF | BF | AF |
| Ours | StanfordDogs | Pekinese | 0.705 | 0.0 | 0.515 | 0.51 | 0.558 | 0.559 | - | - | 0.857 | 0.853 | 0.661 | 0.659 |
| Ours | StanfordDogs | toy poodle | 0.574 | 0.0 | 0.516 | 0.507 | 0.558 | 0.56 | - | - | 0.857 | 0.857 | 0.661 | 0.644 |
| Ours | StanfordDogs | Scotch terrier | 0.5 | 0.0 | 0.517 | 0.509 | 0.558 | 0.543 | - | - | 0.857 | 0.859 | 0.661 | 0.656 |
| Ours | StanfordCars | 2009 Spyker C8 Coupe | 0.262 | 0.0 | 0.559 | 0.56 | - | - | 0.517 | 0.509 | 0.857 | 0.858 | 0.661 | 0.658 |
| Ours | StanfordCars | 2010 Dodge Ram Pickup 3500 Crew Cab | 0.405 | 0.0 | 0.558 | 0.542 | - | - | 0.517 | 0.512 | 0.857 | 0.851 | 0.661 | 0.654 |
| Ours | StanfordCars | 2011 Ford Ranger SuperCab | 0.524 | 0.0 | 0.558 | 0.549 | - | - | 0.517 | 0.509 | 0.857 | 0.856 | 0.661 | 0.658 |
| Ours | Caltech101 | euphonium | 0.789 | 0.0 | 0.858 | 0.861 | 0.558 | 0.561 | 0.517 | 0.512 | - | - | 0.661 | 0.66 |
| Ours | Caltech101 | minaret | 0.826 | 0.0 | 0.857 | 0.857 | 0.558 | 0.557 | 0.517 | 0.519 | - | - | 0.661 | 0.653 |
| Ours | Caltech101 | platypus | 0.9 | 0.0 | 0.857 | 0.86 | 0.558 | 0.56 | 0.517 | 0.507 | - | - | 0.661 | 0.661 |
| Ours | OxfordFlowers | gazania | 0.957 | 0.0 | 0.658 | 0.646 | 0.558 | 0.555 | 0.517 | 0.514 | 0.857 | 0.859 | - | - |
| Ours | OxfordFlowers | tree mallow | 1.0 | 0.0 | 0.658 | 0.646 | 0.558 | 0.56 | 0.517 | 0.514 | 0.857 | 0.856 | - | - |
| Ours | OxfordFlowers | trumpet creeper | 0.588 | 0.0 | 0.661 | 0.661 | 0.558 | 0.56 | 0.517 | 0.516 | 0.857 | 0.861 | - | - |
| Lip | StanfordDogs | Pekinese | 0.705 | 0.066 | 0.515 | 0.514 | 0.655 | 0.559 | - | - | 0.933 | 0.867 | 0.708 | 0.658 |
| Lip | StanfordDogs | toy poodle | 0.574 | 0.033 | 0.516 | 0.518 | 0.655 | 0.559 | - | - | 0.933 | 0.867 | 0.708 | 0.647 |
| Lip | StanfordDogs | Scotch terrier | 0.5 | 0.047 | 0.517 | 0.516 | 0.655 | 0.557 | - | - | 0.933 | 0.865 | 0.708 | 0.66 |
| Lip | StanfordCars | 2009 Spyker C8 Coupe | 0.262 | 0.024 | 0.559 | 0.553 | - | - | 0.591 | 0.518 | 0.933 | 0.865 | 0.708 | 0.66 |
| Lip | StanfordCars | 2010 Dodge Ram Pickup 3500 Crew Cab | 0.405 | 0.143 | 0.558 | 0.544 | - | - | 0.591 | 0.502 | 0.933 | 0.845 | 0.708 | 0.638 |
| Lip | StanfordCars | 2011 Ford Ranger SuperCab | 0.524 | 0.0 | 0.558 | 0.555 | - | - | 0.591 | 0.52 | 0.933 | 0.869 | 0.708 | 0.661 |
| Lip | Caltech101 | euphonium | 0.789 | 0.0 | 0.858 | 0.868 | 0.655 | 0.557 | 0.591 | 0.52 | - | - | 0.708 | 0.658 |
| Lip | Caltech101 | minaret | 0.826 | 0.043 | 0.857 | 0.863 | 0.655 | 0.556 | 0.591 | 0.515 | - | - | 0.708 | 0.661 |
| Lip | Caltech101 | platypus | 0.9 | 0.2 | 0.857 | 0.866 | 0.655 | 0.558 | 0.591 | 0.524 | - | - | 0.708 | 0.653 |
| Lip | OxfordFlowers | gazania | 0.957 | 0.0 | 0.658 | 0.649 | 0.655 | 0.559 | 0.591 | 0.513 | 0.933 | 0.869 | - | - |
| Lip | OxfordFlowers | tree mallow | 1.0 | 0.0 | 0.658 | 0.643 | 0.655 | 0.557 | 0.591 | 0.51 | 0.933 | 0.869 | - | - |
| Lip | OxfordFlowers | trumpet creeper | 0.588 | 0.0 | 0.661 | 0.643 | 0.655 | 0.557 | 0.591 | 0.503 | 0.933 | 0.866 | - | - |
| Emb | StanfordDogs | Pekinese | 0.705 | 0.361 | 0.515 | 0.484 | 0.558 | 0.559 | - | - | 0.857 | 0.84 | 0.661 | 0.633 |
| Emb | StanfordDogs | toy poodle | 0.574 | 0.361 | 0.516 | 0.481 | 0.558 | 0.553 | - | - | 0.857 | 0.832 | 0.661 | 0.613 |
| Emb | StanfordDogs | Scotch terrier | 0.5 | 0.062 | 0.517 | 0.472 | 0.558 | 0.551 | - | - | 0.857 | 0.837 | 0.661 | 0.617 |
| Emb | StanfordCars | 2009 Spyker C8 Coupe | 0.262 | 0.024 | 0.559 | 0.529 | - | - | 0.517 | 0.508 | 0.857 | 0.841 | 0.661 | 0.639 |
| Emb | StanfordCars | 2010 Dodge Ram Pickup 3500 Crew Cab | 0.405 | 0.119 | 0.558 | 0.542 | - | - | 0.517 | 0.512 | 0.857 | 0.857 | 0.661 | 0.654 |
| Emb | StanfordCars | 2011 Ford Ranger SuperCab | 0.524 | 0.119 | 0.558 | 0.539 | - | - | 0.517 | 0.509 | 0.857 | 0.852 | 0.661 | 0.654 |
| Emb | Caltech101 | euphonium | 0.789 | 0.263 | 0.858 | 0.833 | 0.558 | 0.548 | 0.517 | 0.506 | - | - | 0.661 | 0.616 |
| Emb | Caltech101 | minaret | 0.826 | 0.13 | 0.857 | 0.827 | 0.558 | 0.54 | 0.517 | 0.507 | - | - | 0.661 | 0.639 |
| Emb | Caltech101 | platypus | 0.9 | 0.0 | 0.857 | 0.829 | 0.558 | 0.549 | 0.517 | 0.49 | - | - | 0.661 | 0.597 |
| Emb | OxfordFlowers | gazania | 0.957 | 0.739 | 0.658 | 0.632 | 0.558 | 0.551 | 0.517 | 0.503 | 0.857 | 0.849 | - | - |
| Emb | OxfordFlowers | tree mallow | 1.0 | 0.353 | 0.658 | 0.612 | 0.558 | 0.554 | 0.517 | 0.504 | 0.857 | 0.849 | - | - |
| Emb | OxfordFlowers | trumpet creeper | 0.588 | 0.235 | 0.661 | 0.632 | 0.558 | 0.555 | 0.517 | 0.508 | 0.857 | 0.853 | - | - |
| Amns | StanfordDogs | Pekinese | 0.705 | 0.459 | 0.515 | 0.486 | 0.558 | 0.561 | - | - | 0.857 | 0.847 | 0.661 | 0.65 |
| Amns | StanfordDogs | toy poodle | 0.574 | 0.492 | 0.516 | 0.423 | 0.558 | 0.55 | - | - | 0.857 | 0.839 | 0.661 | 0.628 |
| Amns | StanfordDogs | Scotch terrier | 0.5 | 0.031 | 0.517 | 0.488 | 0.558 | 0.559 | - | - | 0.857 | 0.859 | 0.661 | 0.651 |
| Amns | StanfordCars | 2009 Spyker C8 Coupe | 0.262 | 0.143 | 0.559 | 0.516 | - | - | 0.517 | 0.51 | 0.857 | 0.854 | 0.661 | 0.646 |
| Amns | StanfordCars | 2010 Dodge Ram Pickup 3500 Crew Cab | 0.405 | 0.429 | 0.558 | 0.49 | - | - | 0.517 | 0.5 | 0.857 | 0.868 | 0.661 | 0.658 |
| Amns | StanfordCars | 2011 Ford Ranger SuperCab | 0.524 | 0.5 | 0.558 | 0.489 | - | - | 0.517 | 0.507 | 0.857 | 0.868 | 0.661 | 0.656 |
| Amns | Caltech101 | euphonium | 0.789 | 0.316 | 0.858 | 0.856 | 0.558 | 0.557 | 0.517 | 0.519 | - | - | 0.661 | 0.655 |
| Amns | Caltech101 | platypus | 0.9 | 0.5 | 0.857 | 0.832 | 0.558 | 0.555 | 0.517 | 0.495 | - | - | 0.661 | 0.634 |
| Amns | Caltech101 | minaret | 0.826 | 0.174 | 0.857 | 0.813 | 0.558 | 0.546 | 0.517 | 0.493 | - | - | 0.661 | 0.591 |
| Amns | OxfordFlowers | gazania | 0.957 | 0.87 | 0.658 | 0.595 | 0.558 | 0.557 | 0.517 | 0.489 | 0.857 | 0.834 | - | - |
| Amns | OxfordFlowers | tree mallow | 1.0 | 0.0 | 0.658 | 0.598 | 0.558 | 0.511 | 0.517 | 0.476 | 0.857 | 0.843 | - | - |
| Amns | OxfordFlowers | trumpet creeper | 0.588 | 0.294 | 0.661 | 0.584 | 0.558 | 0.554 | 0.517 | 0.494 | 0.857 | 0.828 | - | - |
| EMMN | StanfordDogs | Pekinese | 0.787 | 0.0 | 0.59 | 0.376 | 0.655 | 0.278 | - | - | 0.933 | 0.828 | 0.708 | 0.432 |
| EMMN | StanfordDogs | toy poodle | 0.607 | 0.0 | 0.591 | 0.373 | 0.655 | 0.308 | - | - | 0.933 | 0.836 | 0.708 | 0.446 |
| EMMN | StanfordDogs | Scotch terrier | 0.625 | 0.125 | 0.591 | 0.347 | 0.655 | 0.265 | - | - | 0.933 | 0.813 | 0.708 | 0.436 |
| EMMN | StanfordCars | 2009 Spyker C8 Coupe | 0.429 | 0.0 | 0.656 | 0.188 | - | - | 0.591 | 0.116 | 0.933 | 0.614 | 0.708 | 0.148 |
| EMMN | StanfordCars | 2010 Dodge Ram Pickup 3500 Crew Cab | 0.548 | 0.476 | 0.656 | 0.184 | - | - | 0.591 | 0.13 | 0.933 | 0.56 | 0.708 | 0.126 |
| EMMN | StanfordCars | 2011 Ford Ranger SuperCab | 0.81 | 0.0 | 0.654 | 0.175 | - | - | 0.591 | 0.111 | 0.933 | 0.594 | 0.708 | 0.136 |
| EMMN | Caltech101 | euphonium | 1.0 | 0.105 | 0.933 | 0.783 | 0.655 | 0.352 | 0.591 | 0.297 | - | - | 0.708 | 0.45 |
| EMMN | Caltech101 | minaret | 0.913 | 0.348 | 0.933 | 0.817 | 0.655 | 0.36 | 0.591 | 0.315 | - | - | 0.708 | 0.485 |
| EMMN | Caltech101 | platypus | 1.0 | 0.4 | 0.933 | 0.838 | 0.655 | 0.345 | 0.591 | 0.294 | - | - | 0.708 | 0.484 |
| EMMN | OxfordFlowers | gazania | 1.0 | 0.0 | 0.705 | 0.44 | 0.655 | 0.308 | 0.591 | 0.312 | 0.933 | 0.832 | - | - |
| EMMN | OxfordFlowers | tree mallow | 0.765 | 0.0 | 0.707 | 0.445 | 0.655 | 0.33 | 0.591 | 0.288 | 0.933 | 0.829 | - | - |
| EMMN | OxfordFlowers | trumpet creeper | 0.588 | 0.059 | 0.709 | 0.413 | 0.655 | 0.312 | 0.591 | 0.31 | 0.933 | 0.828 | - | - |

# B ViT-B/16 Full Results

Table 10: Forgetting results with ViT-B/16 visual encoder. We compare our methods with four others on three classes for four selected datasets.

| Method | Dataset | Class name | Target Class acc. | | Other Classes acc. | | StanfordCars | | StanfordDogs | | Caltech101 | | OxfordFlowers | |
|---|---|---|---|---|---|---|---|---|---|---|---|---|---|---|
| | | | BF | AF | BF | AF | BF | AF | BF | AF | BF | AF | BF | AF |
| Ours | StanfordDogs | Pekinese | 0.787 | 0.0 | 0.59 | 0.586 | 0.655 | 0.653 | - | - | 0.933 | 0.933 | 0.708 | 0.692 |
| Ours | StanfordDogs | toy poodle | 0.607 | 0.0 | 0.591 | 0.581 | 0.655 | 0.651 | - | - | 0.933 | 0.932 | 0.708 | 0.7 |
| Ours | StanfordDogs | Scotch terrier | 0.625 | 0.0 | 0.591 | 0.58 | 0.655 | 0.654 | - | - | 0.933 | 0.924 | 0.708 | 0.698 |
| Ours | StanfordCars | 2009 Spyker C8 Coupe | 0.429 | 0.0 | 0.656 | 0.643 | - | - | 0.591 | 0.592 | 0.933 | 0.935 | 0.708 | 0.701 |
| Ours | StanfordCars | 2010 Dodge Ram Pickup 3500 Crew Cab | 0.548 | 0.0 | 0.656 | 0.646 | - | - | 0.591 | 0.591 | 0.933 | 0.932 | 0.708 | 0.703 |
| Ours | StanfordCars | 2011 Ford Ranger SuperCab | 0.81 | 0.0 | 0.654 | 0.639 | - | - | 0.591 | 0.589 | 0.933 | 0.934 | 0.708 | 0.703 |
| Ours | Caltech101 | euphonium | 1.0 | 0.0 | 0.933 | 0.93 | 0.655 | 0.651 | 0.591 | 0.56 | - | - | 0.708 | 0.692 |
| Ours | Caltech101 | minaret | 0.913 | 0.0 | 0.933 | 0.934 | 0.655 | 0.654 | 0.591 | 0.588 | - | - | 0.708 | 0.705 |
| Ours | Caltech101 | platypus | 1.0 | 0.0 | 0.933 | 0.932 | 0.655 | 0.654 | 0.591 | 0.573 | - | - | 0.708 | 0.701 |
| Ours | OxfordFlowers | gazania | 1.0 | 0.0 | 0.705 | 0.702 | 0.655 | 0.652 | 0.591 | 0.583 | 0.933 | 0.932 | - | - |
| Ours | OxfordFlowers | tree mallow | 0.765 | 0.0 | 0.707 | 0.703 | 0.655 | 0.653 | 0.591 | 0.58 | 0.933 | 0.933 | - | - |
| Ours | OxfordFlowers | trumpet creeper | 0.588 | 0.0 | 0.709 | 0.709 | 0.655 | 0.656 | 0.591 | 0.59 | 0.933 | 0.933 | - | - |
| Lip | StanfordDogs | Pekinese | 0.787 | 0.377 | 0.59 | 0.601 | 0.655 | 0.656 | - | - | 0.933 | 0.934 | 0.708 | 0.708 |
| Lip | StanfordDogs | toy poodle | 0.607 | 0.033 | 0.591 | 0.593 | 0.655 | 0.639 | - | - | 0.933 | 0.932 | 0.708 | 0.707 |
| Lip | StanfordDogs | Scotch terrier | 0.625 | 0.016 | 0.591 | 0.582 | 0.655 | 0.647 | - | - | 0.933 | 0.938 | 0.708 | 0.713 |
| Lip | StanfordCars | 2009 Spyker C8 Coupe | 0.429 | 0.262 | 0.656 | 0.639 | - | - | 0.591 | 0.581 | 0.933 | 0.93 | 0.708 | 0.7 |
| Lip | StanfordCars | 2010 Dodge Ram Pickup 3500 Crew Cab | 0.548 | 0.048 | 0.656 | 0.634 | - | - | 0.591 | 0.58 | 0.933 | 0.933 | 0.708 | 0.708 |
| Lip | StanfordCars | 2011 Ford Ranger SuperCab | 0.81 | 0.167 | 0.654 | 0.653 | - | - | 0.591 | 0.59 | 0.933 | 0.933 | 0.708 | 0.713 |
| Lip | Caltech101 | euphonium | 1.0 | 0.158 | 0.933 | 0.935 | 0.655 | 0.653 | 0.591 | 0.597 | - | - | 0.708 | 0.706 |
| Lip | Caltech101 | minaret | 0.913 | 0.87 | 0.933 | 0.932 | 0.655 | 0.649 | 0.591 | 0.59 | - | - | 0.708 | 0.709 |
| Lip | Caltech101 | platypus | 1.0 | 0.7 | 0.933 | 0.936 | 0.655 | 0.653 | 0.591 | 0.595 | - | - | 0.708 | 0.711 |
| Lip | OxfordFlowers | gazania | 1.0 | 0.0 | 0.705 | 0.7 | 0.655 | 0.642 | 0.591 | 0.587 | 0.933 | 0.935 | - | - |
| Lip | OxfordFlowers | tree mallow | 0.765 | 0.176 | 0.707 | 0.699 | 0.655 | 0.65 | 0.591 | 0.596 | 0.933 | 0.933 | - | - |
| Lip | OxfordFlowers | trumpet creeper | 0.588 | 0.059 | 0.709 | 0.705 | 0.655 | 0.644 | 0.591 | 0.581 | 0.933 | 0.932 | - | - |
| Emb | StanfordDogs | Pekinese | 0.787 | 0.213 | 0.59 | 0.601 | 0.655 | 0.656 | - | - | 0.933 | 0.934 | 0.708 | 0.708 |
| Emb | StanfordDogs | toy poodle | 0.607 | 0.0 | 0.591 | 0.472 | 0.655 | 0.621 | - | - | 0.933 | 0.931 | 0.708 | 0.696 |
| Emb | StanfordDogs | Scotch terrier | 0.625 | 0.0 | 0.591 | 0.481 | 0.655 | 0.617 | - | - | 0.933 | 0.926 | 0.708 | 0.695 |
| Emb | StanfordCars | 2009 Spyker C8 Coupe | 0.429 | 0.0 | 0.656 | 0.479 | - | - | 0.591 | 0.392 | 0.933 | 0.908 | 0.708 | 0.659 |
| Emb | StanfordCars | 2010 Dodge Ram Pickup 3500 Crew Cab | 0.548 | 0.0 | 0.656 | 0.626 | - | - | 0.591 | 0.59 | 0.933 | 0.934 | 0.708 | 0.713 |
| Emb | StanfordCars | 2011 Ford Ranger SuperCab | 0.81 | 0.0 | 0.654 | 0.565 | - | - | 0.591 | 0.542 | 0.933 | 0.92 | 0.708 | 0.699 |
| Emb | Caltech101 | euphonium | 1.0 | 0.368 | 0.933 | 0.935 | 0.655 | 0.652 | 0.591 | 0.594 | - | - | 0.708 | 0.709 |
| Emb | Caltech101 | minaret | 0.913 | 0.826 | 0.933 | 0.933 | 0.655 | 0.635 | 0.591 | 0.583 | - | - | 0.708 | 0.711 |
| Emb | Caltech101 | platypus | 1.0 | 0.6 | 0.933 | 0.861 | 0.655 | 0.539 | 0.591 | 0.376 | - | - | 0.708 | 0.547 |
| Emb | OxfordFlowers | gazania | 1.0 | 0.0 | 0.705 | 0.705 | 0.655 | 0.645 | 0.591 | 0.593 | 0.933 | 0.933 | - | - |
| Emb | OxfordFlowers | tree mallow | 0.765 | 0.0 | 0.707 | 0.577 | 0.655 | 0.58 | 0.591 | 0.501 | 0.933 | 0.903 | - | - |
| Emb | OxfordFlowers | trumpet creeper | 0.588 | 0.0 | 0.709 | 0.569 | 0.655 | 0.406 | 0.591 | 0.472 | 0.933 | 0.88 | - | - |
| Amns | StanfordDogs | Pekinese | 0.787 | 0.623 | 0.59 | 0.366 | 0.655 | 0.581 | - | - | 0.933 | 0.896 | 0.708 | 0.609 |
| Amns | StanfordDogs | toy poodle | 0.607 | 0.033 | 0.591 | 0.234 | 0.655 | 0.57 | - | - | 0.933 | 0.899 | 0.708 | 0.482 |
| Amns | StanfordDogs | Scotch terrier | 0.625 | 0.0 | 0.591 | 0.473 | 0.655 | 0.618 | - | - | 0.933 | 0.908 | 0.708 | 0.626 |
| Amns | StanfordCars | 2009 Spyker C8 Coupe | 0.429 | 0.0 | 0.656 | 0.058 | - | - | 0.591 | 0.242 | 0.933 | 0.808 | 0.708 | 0.361 |
| Amns | StanfordCars | 2010 Dodge Ram Pickup 3500 Crew Cab | 0.548 | 0.214 | 0.656 | 0.166 | - | - | 0.591 | 0.436 | 0.933 | 0.904 | 0.708 | 0.572 |
| Amns | StanfordCars | 2011 Ford Ranger SuperCab | 0.81 | 0.214 | 0.654 | 0.315 | - | - | 0.591 | 0.516 | 0.933 | 0.916 | 0.708 | 0.596 |
| Amns | Caltech101 | euphonium | 1.0 | 1.0 | 0.933 | 0.901 | 0.655 | 0.648 | 0.591 | 0.57 | - | - | 0.708 | 0.639 |
| Amns | Caltech101 | minaret | 0.913 | 0.739 | 0.933 | 0.774 | 0.655 | 0.336 | 0.591 | 0.257 | - | - | 0.708 | 0.366 |
| Amns | Caltech101 | platypus | 1.0 | 0.8 | 0.933 | 0.868 | 0.655 | 0.566 | 0.591 | 0.507 | - | - | 0.708 | 0.594 |
| Amns | OxfordFlowers | gazania | 1.0 | 0.913 | 0.705 | 0.518 | 0.655 | 0.586 | 0.591 | 0.514 | 0.933 | 0.908 | - | - |
| Amns | OxfordFlowers | tree mallow | 0.765 | 0.824 | 0.707 | 0.484 | 0.655 | 0.593 | 0.591 | 0.513 | 0.933 | 0.91 | - | - |
| Amns | OxfordFlowers | trumpet creeper | 0.588 | 0.765 | 0.709 | 0.578 | 0.655 | 0.627 | 0.591 | 0.55 | 0.933 | 0.92 | - | - |
| EMMN | StanfordDogs | Pekinese | 0.787 | 0.0 | 0.59 | 0.376 | 0.655 | 0.278 | - | - | 0.933 | 0.828 | 0.708 | 0.432 |
| EMMN | StanfordDogs | toy poodle | 0.607 | 0.0 | 0.591 | 0.373 | 0.655 | 0.308 | - | - | 0.933 | 0.836 | 0.708 | 0.446 |
| EMMN | StanfordDogs | Scotch terrier | 0.625 | 0.125 | 0.591 | 0.347 | 0.655 | 0.265 | - | - | 0.933 | 0.813 | 0.708 | 0.436 |
| EMMN | StanfordCars | 2009 Spyker C8 Coupe | 0.429 | 0.0 | 0.656 | 0.188 | - | - | 0.591 | 0.116 | 0.933 | 0.614 | 0.708 | 0.148 |
| EMMN | StanfordCars | 2010 Dodge Ram Pickup 3500 Crew Cab | 0.548 | 0.476 | 0.656 | 0.184 | - | - | 0.591 | 0.13 | 0.933 | 0.56 | 0.708 | 0.126 |
| EMMN | StanfordCars | 2011 Ford Ranger SuperCab | 0.81 | 0.0 | 0.654 | 0.175 | - | - | 0.591 | 0.111 | 0.933 | 0.594 | 0.708 | 0.136 |
| EMMN | Caltech101 | euphonium | 1.0 | 0.105 | 0.933 | 0.783 | 0.655 | 0.352 | 0.591 | 0.297 | - | - | 0.708 | 0.45 |
| EMMN | Caltech101 | minaret | 0.913 | 0.348 | 0.933 | 0.817 | 0.655 | 0.36 | 0.591 | 0.315 | - | - | 0.708 | 0.485 |
| EMMN | Caltech101 | platypus | 1.0 | 0.4 | 0.933 | 0.838 | 0.655 | 0.345 | 0.591 | 0.294 | - | - | 0.708 | 0.484 |
| EMMN | OxfordFlowers | gazania | 1.0 | 0.0 | 0.705 | 0.44 | 0.655 | 0.308 | 0.591 | 0.312 | 0.933 | 0.832 | - | - |
| EMMN | OxfordFlowers | tree mallow | 0.765 | 0.0 | 0.707 | 0.445 | 0.655 | 0.33 | 0.591 | 0.288 | 0.933 | 0.829 | - | - |
| EMMN | OxfordFlowers | trumpet creeper | 0.588 | 0.059 | 0.709 | 0.413 | 0.655 | 0.312 | 0.591 | 0.31 | 0.933 | 0.828 | - | - |

## C  Retrieval Task

We additionally evaluate CLIP on the retrieval task after class forgetting. Following (Kravets & Namboodiri, 2025), we evaluate retrieval of image from text input. We evaluate retrieval creating a database from the four datasets we used in our main experiments. We use precision@k metric for k of 1, 5 and 10, which measures the proportion of relevant items among the top K retrieved results. Lower precision@k indicates better performance. These results are displayed in Tab 11 where we compare the *original*, *Lip* and our method aggregating across all the classes and datasets. Our method achieves best performance also on the retrieval task. Full results are presented in Tabs. 13 and 14.

Table 11: Aggregated across all datasets and classes image retrieval from text input results showing precision@k for k of 1, 5 and 10 with RN50 and ViT-B/16 visual encoders.

| Model | Precision@1 ($\downarrow$) | Precision@5 ($\downarrow$) | Precision@10 ($\downarrow$) |
|---|---|---|---|
| RN50 (original) | 0.833 | 0.683 | 0.583 |
| RN50 (Lip) | 0.08 | 0.23 | 0.191 |
| RN50 (Ours) | **0.0** | **0.017** | **0.008** |
| ViT-B/16 (original) | 0.833 | 0.717 | 0.667 |
| ViT-B/16 (Lip) | 0.5 | 0.433 | 0.4 |
| ViT-B/16 (Ours) | **0.0** | **0.0** | **0.0** |

We would like to note that as we do not modify the visual encoder, the effect of the classes to forget still exists there. Thus, retrieval of images with an input image could be still achieved. However, as demonstrated in Kravets & Namboodiri (2025), even if uni-modal information still exists in the model, breaking the multi-modal link is enough for class forgetting. This is because during image retrieval, a strong model like CLIP can still identify similar features and shapes of objects *without actually recognizing or knowing the specific class they belong to*. This can be seen when performing image-to-image retrieval using the original CLIP model on classes where its classification accuracy is zero. The results presented in Tab. 12 from Kravets & Namboodiri (2025) support this. Therefore, we can conclude that breaking the text-image link is sufficient *specifically* for achieving class forgetting.

Table 12: Image retrieval from image input results on classes with zero classification accuracy using the **original** model.

| Model Type | Class | Classification Accuracy | Precision@1 | Precision@5 | Precision@10 |
|---|---|---|---|---|---|
| CLIP original | Appenzeller (StanfordDogs) | 0 | 1.0 | 0.4 | 0.2 |
| CLIP original | Pembroke (StanfordDogs) | 0 | 1.0 | 0.6 | 0.4 |
| CLIP original | Cardigan (StanfordDogs) | 0 | 0.0 | 0.2 | 0.2 |
| CLIP original | 2010 Chevrolet HHR SS (StanfordCars) | 0 | 1.0 | 0.4 | 0.4 |
| CLIP original | 2009 HUMMER H2 SUT Crew Cab (StanfordCars) | 0 | 1.0 | 0.6 | 0.7 |
| CLIP original | english marigold (OxfordFlowers) | 0 | 1.0 | 0.8 | 0.6 |
| CLIP original | colt's foot (OxfordFlowers) | 0 | 1.0 | 0.8 | 0.7 |
| CLIP original | cape flower (OxfordFlowers) | 0 | 1.0 | 1.0 | 1.0 |

Table 13: Image retrieval from text input results showing precision@k for k of 1, 5 and 10 using RN50 model

| Model Type | Class | Precision@1 | Precision@5 | Precision@10 |
|---|---|---|---|---|
| CLIP original | Scotch terrier | 1.0 | 0.2 | 0.2 |
| CLIP original | toy poodle | 1.0 | 0.6 | 0.5 |
| CLIP original | Pekinese | 1.0 | 0.8 | 0.6 |
| CLIP original | 2009 Spyker C8 Coupe | 1.0 | 0.6 | 0.5 |
| CLIP original | 2010 Dodge Ram Pickup 3500 Crew Cab | 1.0 | 0.2 | 0.2 |
| CLIP original | 2011 Ford Ranger SuperCab | 0.0 | 0.2 | 0.2 |
| CLIP original | euphonium | 1.0 | 1.0 | 1.0 |
| CLIP original | minaret | 1.0 | 1.0 | 1.0 |
| CLIP original | platypus | 1.0 | 1.0 | 0.6 |
| CLIP original | gazania | 1.0 | 1.0 | 1.0 |
| CLIP original | tree mallow | 0.0 | 0.8 | 0.7 |
| CLIP original | trumpet creeper | 1.0 | 0.8 | 0.5 |
| CLIP original Mean | - | 0.833 | 0.683 | 0.583 |
| CLIP forget (Lip) | Scotch terrier | 0.0 | 0.0 | 0.0 |
| CLIP forget (Lip) | toy poodle | 1.0 | 0.2 | 0.1 |
| CLIP forget (Lip) | Pekinese | 0.0 | 0.0 | 0.0 |
| CLIP forget (Lip) | 2009 Spyker C8 Coupe | 0.0 | 0.8 | 0.5 |
| CLIP forget (Lip) | 2010 Dodge Ram Pickup 3500 Crew Cab | 0.0 | 0.2 | 0.3 |
| CLIP forget (Lip) | 2011 Ford Ranger SuperCab | 0.0 | 0.0 | 0.0 |
| CLIP forget (Lip) | euphonium | 0.0 | 0.8 | 0.8 |
| CLIP forget (Lip) | minaret | 0.0 | 0.4 | 0.2 |
| CLIP forget (Lip) | platypus | 0.0 | 0.2 | 0.2 |
| CLIP forget (Lip) | gazania | 0.0 | 0.0 | 0.0 |
| CLIP forget (Lip) | tree mallow | 0.0 | 0.2 | 0.2 |
| CLIP forget (Lip) | trumpet creeper | 0.0 | 0.0 | 0.0 |
| CLIP forget Mean (Lip) | - | 0.08 | 0.23 | 0.191 |
| CLIP forget (Ours) | Scotch terrier | 0.0 | 0.0 | 0.0 |
| CLIP forget (Ours) | toy poodle | 0.0 | 0.0 | 0.0 |
| CLIP forget (Ours) | Pekinese | 0.0 | 0.0 | 0.0 |
| CLIP forget (Ours) | 2009 Spyker C8 Coupe | 0.0 | 0.0 | 0.0 |
| CLIP forget (Ours) | 2010 Dodge Ram Pickup 3500 Crew Cab | 0.0 | 0.0 | 0.0 |
| CLIP forget (Ours) | 2011 Ford Ranger SuperCab | 0.0 | 0.2 | 0.1 |
| CLIP forget (Ours) | euphonium | 0.0 | 0.0 | 0.0 |
| CLIP forget (Ours) | minaret | 0.0 | 0.0 | 0.0 |
| CLIP forget (Ours) | platypus | 0.0 | 0.0 | 0.0 |
| CLIP forget (Ours) | gazania | 0.0 | 0.0 | 0.0 |
| CLIP forget (Ours) | tree mallow | 0.0 | 0.0 | 0.0 |
| CLIP forget (Ours) | trumpet creeper | 0.0 | 0.0 | 0.0 |
| CLIP forget Mean (Ours) | - | **0.0** | **0.017** | **0.008** |

Table 14: Image retrieval from text input results showing precision@k for k of 1, 5 and 10 using ViT-B/16 model

| Model Type | Class | Precision@1 | Precision@5 | Precision@10 |
|---|---|---|---|---|
| CLIP original | Scotch terrier | 0.0 | 0.0 | 0.1 |
| CLIP original | toy poodle | 1.0 | 0.8 | 0.7 |
| CLIP original | Pekinese | 1.0 | 0.4 | 0.5 |
| CLIP original | 2009 Spyker C8 Coupe | 1.0 | 0.8 | 0.8 |
| CLIP original | 2010 Dodge Ram Pickup 3500 Crew Cab | 1.0 | 0.6 | 0.5 |
| CLIP original | 2011 Ford Ranger SuperCab | 1.0 | 0.8 | 0.5 |
| CLIP original | euphonium | 1.0 | 1.0 | 1.0 |
| CLIP original | minaret | 1.0 | 1.0 | 1.0 |
| CLIP original | platypus | 1.0 | 1.0 | 0.9 |
| CLIP original | gazania | 1.0 | 1.0 | 1.0 |
| CLIP original | tree mallow | 0.0 | 0.4 | 0.4 |
| CLIP original | trumpet creeper | 1.0 | 0.8 | 0.6 |
| CLIP original Mean | - | 0.833 | 0.717 | 0.667 |
| CLIP forget (Lip) | Scotch terrier | 0.0 | 0.4 | 0.4 |
| CLIP forget (Lip) | toy poodle | 0.0 | 0.0 | 0.1 |
| CLIP forget (Lip) | Pekinese | 0.0 | 0.0 | 0.2 |
| CLIP forget (Lip) | 2009 Spyker C8 Coupe | 1.0 | 0.8 | 0.8 |
| CLIP forget (Lip) | 2010 Dodge Ram Pickup 3500 Crew Cab | 0.0 | 0.0 | 0.1 |
| CLIP forget (Lip) | 2011 Ford Ranger SuperCab | 1.0 | 0.6 | 0.4 |
| CLIP forget (Lip) | euphonium | 1.0 | 1.0 | 0.6 |
| CLIP forget (Lip) | minaret | 1.0 | 1.0 | 0.9 |
| CLIP forget (Lip) | platypus | 1.0 | 1.0 | 0.5 |
| CLIP forget (Lip) | gazania | 1.0 | 0.2 | 0.4 |
| CLIP forget (Lip) | tree mallow | 0.0 | 0.0 | 0.2 |
| CLIP forget (Lip) | trumpet creeper | 0.0 | 0.2 | 0.2 |
| CLIP forget Mean (Lip) | - | 0.5 | 0.433 | 0.4 |
| CLIP forget (Ours) | Scotch terrier | 0.0 | 0.0 | 0.0 |
| CLIP forget (Ours) | toy poodle | 0.0 | 0.0 | 0.0 |
| CLIP forget (Ours) | Pekinese | 0.0 | 0.0 | 0.0 |
| CLIP forget (Ours) | 2009 Spyker C8 Coupe | 0.0 | 0.0 | 0.0 |
| CLIP forget (Ours) | 2010 Dodge Ram Pickup 3500 Crew Cab | 0.0 | 0.0 | 0.0 |
| CLIP forget (Ours) | 2011 Ford Ranger SuperCab | 0.0 | 0.0 | 0.0 |
| CLIP forget (Ours) | euphonium | 0.0 | 0.0 | 0.0 |
| CLIP forget (Ours) | minaret | 0.0 | 0.0 | 0.0 |
| CLIP forget (Ours) | platypus | 0.0 | 0.0 | 0.0 |
| CLIP forget (Ours) | gazania | 0.0 | 0.0 | 0.0 |
| CLIP forget (Ours) | tree mallow | 0.0 | 0.0 | 0.0 |
| CLIP forget (Ours) | trumpet creeper | 0.0 | 0.0 | 0.0 |
| CLIP forget Mean (Ours) | - | **0.0** | **0.0** | **0.0** |

# D  Interpreting Difficulty of Forgetting a Class

We can directly analyze the low-rank adaptation change in the projection matrix to understand the difficulty of forgetting a certain class. Specifically, we examine the Frobenius norm of the adaptation matrix. Our hypothesis is that a larger Frobenius norm means that a greater modification in the projection matrix is required to successfully forget a target class. Such greater modification will make maintaining other classes accuracy on the similar level harder and thus the average score metrics (the lower the better) will increase as well.

The Frobenius norm of a matrix A is defined as:

$$\|A\|_F = \sqrt{\sum_{i,j} |a_{ij}|^2} \tag{5}$$

We now plot the Frobenius norm of the projection change matrix alongside the average score metrics for the two networks. To ensure more robust statistical results, we include 30 randomly sampled classes:

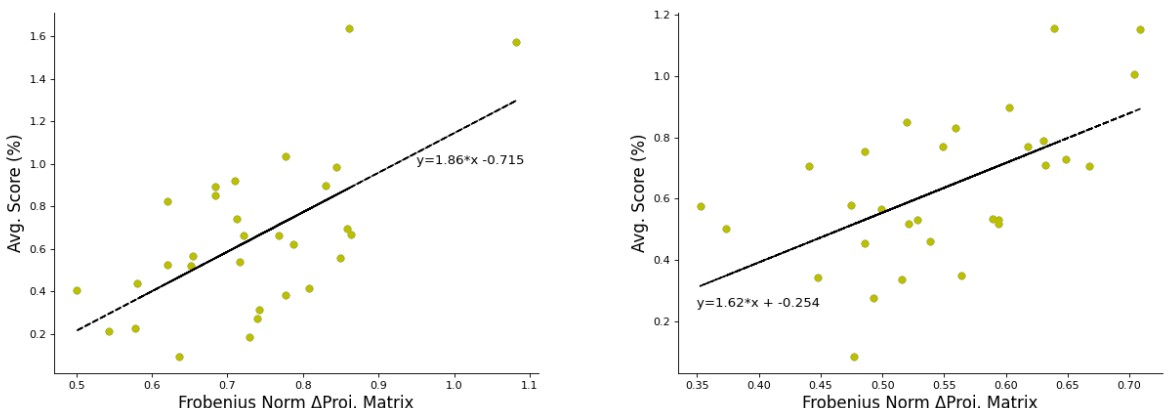

Figure 5: Interpreting the difficulty of forgetting a class by looking at the Frobenius norm of the projection change matrix. Figure on the **left** shows the results for the RN50 model and on the **right** for the ViT-B/16 model.

In Fig. 5 we observe that there is a positive relation between the Frobenius norm of the change in the projection matrix and the average score metrics confirming our hypothesis. The correlation between the Frobenius norm of the change is 0.62 and 0.59 respectively for RN50 and ViT-B/16 models showing that the relation between the two is significant.

# E  Additional Considerations

**Is considering one downstream task enough for forgetting a class?**   We evaluated the performance of our model after forgetting the "Pekinese" class using the ImageNet dataset, employing a ViT-B/16 backbone for these experiments. We observed that the model's accuracy after forgetting is very low on the class. Additionally, we tested the model on a set of 20 randomly sourced "Pekinese" images from the web and found that forgetting was effective in this context as well. This indicates that our approach successfully generalizes beyond the StanfordDogs dataset.

It's important to highlight that our method focuses exclusively on modifying the text space without altering the image encoding space. Images from StanfordDogs, which exhibit diverse backgrounds and poses, cover a wide range of scenarios where the "Pekinese" class might appear. This diversity suggests that testing on a single downstream dataset can be sufficient for evaluation, provided a large number of samples are used.

**Can simply excluding a class from the set of predicted outputs, or using a conditional statement to alter the output if a specific class is detected be considered as class forgetting?** Removing a specific class from a model's classification output is a viable solution when the model is deployed via e.g. an API where the user lacks access to the underlying weights. However, if the model's weights and code are publicly available, such solution could be easily circumvented by the user. In contrast, our method, which directly alters the weight matrices of CLIP, ensures a more robust form of class forgetting. Once the relevant information is erased from the model, it becomes challenging for a user to restore it without extensive fine-tuning on the removed class. This process would be particularly difficult if the data for that class (e.g., a specific individual's photos) is scarce or unavailable online, effectively making the erased class irrecoverable.

## F   Implementation Details

We ran experiments using two versions of CLIP where either ResNet50 or ViT-B/16 [2] visual encoders were used. For both the models we use the $\lambda_1$ of 0.3, $\lambda_3$ of 1 and a varying $\lambda_2$ with initial value of 1.1 incrementing by 0.05 until the reduction in the second loss component exceeds 0.75% of its initial value. We optimize the low-ranking matrices $A$ and $B$ of rank $r$ of 5 for 2000 iterations using Adam optimizer with learning rate of 0.01 and saving the weights that achieve the minimum loss. We use an empty template with only the name of the class when forgetting.

## G   Sequential Multi-class Forgetting

In the main paper we demonstrated multi-class forgetting in scenarios where requests to forget encompass all classes simultaneously. Here, we extend this analysis to scenarios where requests for class forgetting arrive sequentially at different time points. We first forget the first class (in *Classes* column), e.g. "Pekinese" in *StanfordDogs* dataset. Subsequently, we forget two additional classes ("toy poodle" and "Scotch terrier"), starting from the model state that has already forgotten the first class. As presented in Table 15, the forgetting mechanism remains effective even under sequential class forgetting, where requests are received at different intervals.

Table 15: Forgetting on multiple classes with sequential forgetting requests.

| Method | Model | Dataset | Classes | Avg. Target Classes acc. | | Other Classes acc. | | StanfordCars | | StanfordDogs | | Caltech101 | | OxfordFlowers | | Avg. Score ($\downarrow$) |
|---|---|---|---|---|---|---|---|---|---|---|---|---|---|---|---|---|
| | | | | BF | AF | BF | AF | BF | AF | BF | AF | BF | AF | BF | AF | |
| Ours | RN50 | StanfordDogs | Pekinese,toy poodle,Scotch terrier | 0.591 | 0.011 | 0.515 | 0.498 | 0.558 | 0.542 | - | - | 0.857 | 0.854 | 0.661 | 0.633 | 0.025 |
| Ours | RN50 | StanfordCars | 2009 Spyker C8 Coupe, 2010 Dodge Ram Pickup 3500 Crew Cab, 2011 Ford Ranger SuperCab | 0.397 | 0.0 | 0.56 | 0.528 | - | - | 0.517 | 0.499 | 0.857 | 0.847 | 0.661 | 0.653 | 0.023 |
| Ours | RN50 | Caltech101 | euphonium,minaret,platypus | 0.827 | 0.0 | 0.858 | 0.864 | 0.558 | 0.55 | 0.517 | 0.498 | - | - | 0.661 | 0.656 | 0.012 |
| Ours | RN50 | OxfordFlowers | trumpet creeper,gazania,tree mallow | 0.86 | 0.0 | 0.656 | 0.626 | 0.558 | 0.555 | 0.517 | 0.502 | 0.857 | 0.857 | - | - | 0.016 |

## H   Detailed Example on Magic Neurons

In this section we provide some additional clarification on the magic neurons using CLIP with RN50 visual encoder. Assume that the class to be forgotten is "Scotch Terrier", feature 222 undergoes substantial changes after passing through the modified text projection matrix increasing the value of the feature 222 significantly so that it is pushed towards 0 and the dot product with the image embedding for that class becomes small. However, for other classes, feature 222 remains relatively unchanged, so the dot product remains high, allowing these classes to be accurately classified. In Tab. 16 we can observe that the value of feature 222 changes significantly for "Scotch Terrier" between the model before forgetting and after forgetting, while it remains stable for other classes. This ensures that the forgetting process only affects the targeted class without disrupting the other classes.

---

[2]Weigths from https://github.com/openai/CLIP/blob/main/clip/clip.py#L30

Table 16: Values of the feature 222 for different breeds, with Scotch Terrier being the forgotten class.

| Class | Model AF | Model BF |
|---|---|---|
| Scotch Terrier (Forgotten class) | -0.9342 | -4.1981 |
| Toy Poodle | -6.5257 | -6.5185 |
| Pekinese | -4.3343 | -3.9811 |
| Spaniel | -6.5226 | -6.8187 |

## I  Additional Evaluation of Class Forgetting

We evaluate the RN50 CLIP model before forgetting and CLIP model with "Scotch terrier" class forgotten on the validation sets of the MS-COCO and ImageNet datasets. MS-COCO is a multi-label classification dataset with dense, complex images, while ImageNet is a highly diverse dataset spanning a wide range of classes.

The results presented in Tab. 17 demonstrate that the accuracy of unaffected classes on these datasets does not change after forgetting. This stability holds across both the multi-label classification setting of MS-COCO and the highly varied class distribution of ImageNet, validating the effectiveness of our method for targeted class forgetting.

| Dataset | CLIP After Forgetting | CLIP Before Forgetting |
|---|---|---|
| MS-COCO (all classes) | 0.9661 | 0.9667 |
| ImageNet (all classes) | 0.5756 | 0.5794 |

Table 17: Comparison Accuracy of CLIP before forgetting vs CLIP after forgetting on MS-Coco and ImageNet.

## J  Generated Semantically Similar Classes to Preserve

Here we show the list of semantically similar classes generated by an LLM with a prompt *Generate semantically similar classes to {class}*.

**StanfordDogs**:
Shih Tzu, Lhasa Apso, Maltese, Havanese, Bichon Frise, Yorkshire Terrier, Pomeranian, Cavalier King Charles Spaniel, Papillon, Japanese Chin, Brussels Griffon, Miniature Schnauzer, West Highland White Terrier, Cairn Terrier, Norfolk Terrier, Norwich Terrier, Tibetan Spaniel, Tibetan Terrier, Silky Terrier, Affenpinscher, Chinese Crested, Italian Greyhound, Toy Manchester Terrier, Toy Fox Terrier, Australian Terrier, Border Terrier, Dandie Dinmont Terrier, Sealyham Terrier, Skye Terrier, Welsh Terrier, Lakeland Terrier, Jack Russell Terrier, Parson Russell Terrier, Rat Terrier, Bedlington Terrier, Manchester Terrier, Fox Terrier, Wire Fox Terrier, Smooth Fox Terrier, Irish Terrier, Glen of Imaal Terrier, Kerry Blue Terrier, Soft Coated Wheaten Terrier, Bull Terrier, Miniature Bull Terrier, Boston Terrier, French Bulldog, English Bulldog, American Bulldog, Boxer, Pug, Miniature Pinscher, German Pinscher, Doberman Pinscher, Great Dane, Mastiff, Bullmastiff, Neapolitan Mastiff, Dogue de Bordeaux, Rottweiler, Saint Bernard, Bernese Mountain Dog, Greater Swiss Mountain Dog, Newfoundland, Leonberger, Tibetan Mastiff, Chihuahua, Poodle, Miniature Poodle, Standard Poodle, Shetland Sheepdog, Collie, Border Collie, Australian Shepherd, Australian Cattle Dog, Old English Sheepdog, Bearded Collie, Briard, Welsh Corgi, Cardigan Welsh Corgi, Pembroke Welsh Corgi, American Eskimo Dog, Alaskan Malamute, Siberian Husky, Samoyed, Shiba Inu, Akita, Basenji, Beagle, Bloodhound, Basset Hound, Dachshund, Coonhound, Foxhound, Whippet, Greyhound, Saluki, Afghan Hound, Borzoi, Irish Wolfhound, Scottish Deerhound

**StanfordCars**:
Chevrolet Silverado 1500,GMC Sierra 1500,Toyota Tundra,Nissan Titan,Ram 1500,Ford F-150,Honda Ridgeline,Chevrolet Colorado,GMC Canyon,Toyota Tacoma,Nissan Frontier,Jeep Gladiator,Ford Maverick,Hyundai Santa Cruz,Chevrolet Silverado 2500HD,GMC Sierra 2500HD,Ford F-250 Super Duty,Ram 2500,Chevrolet Silverado 3500HD,GMC Sierra 3500HD,Ford F-350 Super Duty,Ram 3500,Chevrolet Silverado 4500HD,Ford F-450 Super Duty,GMC Sierra 4500HD,Ram 4500,Chevrolet

Silverado 5500HD,Ford F-550 Super Duty,GMC Sierra 5500HD,Ram 5500,Ford F-650,Ford
F-750,International CV Series,Mitsubishi Fuso Canter,Isuzu N-Series,Hino 268,Freightliner
M2 106,Peterbilt 220,Kenworth T270,Ram ProMaster,Ford Transit,Mercedes-Benz Sprinter,Nissan
NV,Chevrolet Express,GMC Savana,Ram ProMaster City,Ford Transit Connect,Nissan
NV200,Chevrolet Colorado ZR2,Toyota Tacoma TRD Pro,Jeep Wrangler Rubicon,Ford Ranger
Tremor,Ram Rebel,Chevrolet Silverado Trail Boss,GMC Sierra AT4,Ford F-150 Raptor,Nissan
Titan XD,Toyota Tundra TRD Pro,Chevrolet Avalanche,Honda Element,Ford Explorer Sport
Trac,Lincoln Mark LT,Cadillac Escalade EXT,Hummer H2 SUT,Chevrolet SSR,Subaru Baja,Dodge
Dakota,Mazda B-Series,Mitsubishi Raider,Suzuki Equator,Isuzu i-Series,Ford Courier,Volkswagen
Amarok,Peugeot Landtrek,Fiat Fullback,Renault Alaskan,Mercedes-Benz X-Class,SsangYong
Musso,Great Wall Steed,Mahindra Scorpio Getaway,Tata Xenon,Holden Colorado,HSV Maloo,Ford
Falcon Ute,Chevrolet S-10,Ford Ranger Raptor,RAM 1200,Toyota Hilux,Chevrolet LUV,Ford
Courier,Mazda BT-50,Mitsubishi Triton,Nissan Navara,Isuzu D-Max,Volkswagen Tarok,Jeep
Comanche

**Caltech101**:
Accordion, Bagpipes, Banjo, Bassoon, Cello, Clarinet, Cornet, Double Bass, Drum Set, Flute,
French Horn, Guitar, Harp, Mandolin, Marimba, Oboe, Piano, Saxophone, Sitar, Sousaphone,
Tambourine, Trombone, Trumpet, Tuba, Ukulele, Viola, Violin, Xylophone, Zenko Drum,
Glockenspiel, Concertina, Hurdy-Gurdy, Lute, Melodica, Piccolo, Pipe Organ, Recorder,
Theremin, Triangle, Bass Drum, Cabasa, Castanets, Claves, Conga Drum, Cowbell, Djembe,
Guiro, Kalimba, Maracas, Shekere, Sleigh Bells, Snare Drum, Talking Drum, Timpani, Vibraslap,
Whip, Washboard, Zephyr Organ, Zither, Azimuth Marker, Bell Tower, Belfry, Cathedral Spire,
Church Steeple, Dome, Gazebo, Lighthouse, Obelisk, Pagoda, Watchtower, Water Tower, Windmill,
Cairn, Cenotaph, Column, Monolith, Obelisk, Pavilion, Pyramid, Stupa, Totem Pole, Triumphal
Arch, Rotunda, Spire, Tower, Ziggurat, Amphibian, Anteater, Armadillo, Barramundi, Basilisk,
Beaver, Capybara, Chameleon, Coatimundi, Echidna, Gecko, Gila Monster, Iguana, Komodo Dragon,
Koala, Marsupial, Mole, Monotreme, Newt, Numbat, Opossum, Pangolin, Platypus, Quokka, Quoll,
Salamander, Shrew, Skink, Sloth, Sugar Glider, Tasmanian Devil, Tree Frog, Tuatara, Wombat,
Anhinga, Auk, Bittern, Booby, Cormorant, Crane, Curlew, Egret, Flamingo, Frigatebird, Gannet,
Grebe, Heron, Ibis, Jaeger, Kestrel, Kingfisher, Kittiwake, Loon, Oystercatcher, Pelican,
Petrel, Puffin, Rail, Razorbill, Sandpiper, Shearwater, Skua, Snipe, Tern, Turnstone, Wader,
Whimbrel, Woodcock, Meerkat, Mongoose, Pangolin, Platypus, Potto, Puffin, Quokka, Quoll,
Raccoon, Red Panda, Ringtail, Skunk, Sloth, Sugar Glider, Tasmanian Devil, Tenrec, Tree
Shrew, Wombat, Zebra Finch, Zebu, Zonkey, Zorilla, Zygodont

**OxfordFlowers**:
Rose, Tulip, Lily, Daisy, Sunflower, Orchid, Marigold, Lavender, Daffodil, Chrysanthemum,
Carnation, Hibiscus, Iris, Peony, Poppy, Lotus, Bluebell, Magnolia, Gardenia, Jasmine,
Azalea, Camellia, Geranium, Hyacinth, Petunia, Zinnia, Begonia, Cosmos, Foxglove, Freesia,
Gladiolus, Hollyhock, Lilac, Narcissus, Snapdragon, Sweet Pea, Verbena, Violet, Wisteria,
Aster, Anemone, Gaura, Bachelor's Button, Bellflower, Buttercup, Calla Lily, Canna,
Protea, Columbine, Coreopsis, Delphinium, Gaillardia, Primula, Heliotrope, Impatiens,
Kalanchoe, Lantana, Morning Glory, Nasturtium, Pansy, Phlox, Plumeria, Primrose, Ranunculus,
Rhododendron, Scabiosa, Sedum, Stock, Tithonia, Trillium, Tuberose, Wallflower, Yarrow,
Yucca, Amaryllis, Bougainvillea, Bromelia, Angelonia, Armeria, Balloon Flower, Ballmoss,
Bee Balm, Black-eyed Susan, Bleeding Heart, Borage, Browallia, Candytuft, Clematis, Cleome,
Cockscomb, Coral Bells, Corydalis, Crocosmia, Cyclamen, Diascia, Dusty Miller, Echinacea,
Euphorbia, Four O'Clock, Gazania, Geum

