# OpenReview forum: "Zero-shot CLIP Class Forgetting via Text-image Space Adaptation"
_TMLR — Accepted by TMLR_

### Review · Reviewer_Mfz1 · 2024-09-22

**Summary Of Contributions:**

The paper studies class-wise unlearning for CLIP models without retraining the model with additional data. The main method applies LoRA to optimize the projection of text embedding to the image-text space (which is simply a linear layer) such that the text embedding matches an empty token, thus breaking the connection between the image and text embeddings of CLIP. In the meantime, such projection should also preserve performance in other classes. Experiments show that the proposed method performs well across four datasets.

**Audience:**

Yes

**Claims And Evidence:**

Yes

**Requested Changes:**

1. Consider rewording the entire paper, for example, raise the method as a concept removal paper rather than an unlearning paper;
2. Consider adding a experiment to examine whether the proposed method can remove a class for several downstream datasets;
3. Add the missing details mentioned above;
4. Refine the writing according to the suggestions above.

**Strengths And Weaknesses:**

## Strengths

(1) Unlearning CLIP is an interesting and important task;

(2) Using LoRA for fine-tuning certain weights to achieve unlearning is novel.

## Weaknesses

(1) **Should the task be considered as unlearning?** My biggest concern is that raising the method as an unlearning algorithm may be inaccurate. As the authors described in the introduction, unlearning aims to remove the influence of certain training data. There are two concerns in the CLIP scenario:
- Training data is not available, thus the selection of the unlearn set can be ambiguous, namely that we are not even sure if the data is in the original training set;
- The author mentions that they want the model to perform as badly as possible on the unlearn set while performing as well as possible on the retained set. This does not necessarily align with the goal of unlearning. For example, if the model generalizes well, it can still perform well on unseen data. That is the reason why we normally set the goal (at least one of the goals) to be "the model performance should be roughly the same between the unlearn set and the test set".

As a result, I believe the task of this paper is actually "forgetting a class" or concept removal, which can be slightly different from unlearning. Of course, I am happy to chat more during the rebuttal on the precision of such wording.

(2) **Is considering one downstream task enough for forgetting a class?** Following my previous point, if one wants to remove a class, examining one downstream task may not be enough. In other words, does the proposed method achieve good performance on the same class across different datasets? For example, if I want to remove the class "dog" from CLIP, does your method remove the dogs from both ImageNet and CIFAR-10?

(3) **Missing details** There are many crucial details missing in this paper, making this paper potentially hard to read:
- CLIP is a contrastive learning method, not a specific model. The authors should explicitly present their choice of CLIP model, including the architecture and link to the pre-trained weights they used.
- Section 3 is way too brief and does not include any background information on CLIP or LoRA. The authors should add the mechanism of CLIP, introduce the projection layer (as this is the main focus of the study), and discuss the LoRA algorithm.
- Equation 2 is incomplete, namely, it does not specify which component to optimize or whether you are performing minimization or maximization.

(4) **Writing** Writing can be further refined:
- Citations should be used properly (e.g., \citet and \citep);
- Grammar mistakes such as the sentence below equation (2);
- Figure 1 does not accurately reflect the unlearning procedure;
- Equation 1 does not suggest the right unlearning procedure;

(5) **Finally,** it seems that there exists a super simple trick that can achieve the same goal as the proposed algorithm: I can just add a condition to the model such that if a certain class is detected, the model always outputs a wrong prediction. Can you discuss why this method is not desired?

---

> ### Author Response · Authors · 2024-10-27
> **Author's Response to Reviewer Mfz1 - Part 1**
>
> We appreciate the reviewer for their time and constructive feedback. In response to the reviewer’s suggestions we have revised the paper accordingly and uploaded the updated manuscript with changes highlighted in blue. Below, we address the concerns raised by the reviewer.
>
> **W1: Should the task be considered as unlearning?**
>
> We agree with the reviewer and we rephrased the paper putting it as class/concept forgetting rather than unlearning as we don't use training data. However, as classes can be removed also through machine unlearning in CLIP for example generating synthetic data rather than using real data (that are unknown for CLIP) we also talk about machine unlearning which is still related to concept removal as they have some common goals, in our case breaking the association between visual and textual embedding for a targeted class. We also removed the general formula for machine unlearning as we now rephrased the paper as class removal.
>
>
> **W2: Is considering one downstream task enough for forgetting a class?**
>
> We tested how the unlearned model for "Pekinese" class performs on *ImageNet* (actually it's misspelled "Pekingese" in *ImageNet* ). We see that the model (we used ViT-B/16 backbone for these experiments) after forgetting has the accuracy of 0.01 after class forgetting while it has 0.84 accuracy before forgetting the class. We also take some "Pekinese" classes in the wild downloading 20 random images from Web of "Pekinese" and we see that unlearning has worked for them too. Thus the method worked across *StanfordDogs*, *ImageNet* datasets and random Web samples.
> We would also like to point out that as we are not using or modifying the image encoding space but only the text space, and as for example "Pekinese" images found in *StanfordDogs*  will have different backgrounds, different positions etc so they will cover a lot of situations where "Pekinese" is present. Thus, it's not clear why it wouldn't be working on other datasets that still have images of "Pekinese" but in slightly different positions/on different backgrounds? So we believe showing it on *StanfordDogs* is a sign that the model is not able to recognize "Pekinese" anymore no matter the downstream dataset we choose.
>
> **W3: Missing details There are many crucial details missing in this paper, making this paper potentially hard to read: CLIP is a contrastive learning method, not a specific model. The authors should explicitly present their choice of CLIP model, including the architecture and link to the pre-trained weights they used.
> Section 3 is way too brief and does not include any background information on CLIP or LoRA. The authors should add the mechanism of CLIP, introduce the projection layer (as this is the main focus of the study), and discuss the LoRA algorithm.
> Equation 2 is incomplete, namely, it does not specify which component to optimize or whether you are performing minimization or maximization.**
>
> We added in the footnote that we are using the OpenAI CLIP model with weights from [https://github.com/openai/CLIP/blob/main/clip/clip.py#L30](https://github.com/openai/CLIP/blob/main/clip/clip.py#L30). We added to Section 3 more information about CLIP and LoRA in the revised manuscript. We also added to the loss equation in Section 4 that we optimize the low-rank update matrices A and B and that we minimize the loss. Please let us know if the modifications suffice. We also intend to provide access to the complete implementation with a Google Colab notebook for easy reproducibility of our method on acceptance of the paper.
>
>
> **W4: Writing can be further refined: Citations should be used properly (e.g., citet and citep);
> Grammar mistakes such as the sentence below equation (2);
> Figure 1 does not accurately reflect the unlearning procedure;
> Equation 1 does not suggest the right unlearning procedure;**
>
> We corrected grammar and citations and removed equation 1 as we now rephrased the paper from unlearning to class forgetting. We changed Figure 1 and we hope that it is more reflective of the forgetting procedure.

---

> > ### Author Response · Authors · 2024-10-27
> > **Author's Response to Reviewer Mfz1 - Part 2**
> >
> > **W5: Finally, it seems that there exists a super simple trick that can achieve the same goal as the proposed algorithm: I can just add a condition to the model such that if a certain class is detected, the model always outputs a wrong prediction. Can you discuss why this method is not desired?**
> >
> > This simple trick would work if the model was used via an API for example where the user wouldn't have the access to the weights. Indeed, another "simple" solution would be to remove from the possible classes the unwanted class. However, if the weights and the code to the model are open sourced those "protection layers" such as an if statement could be easily removed by the user. On the other hand, modifying the weights of CLIP it would be hard for the user to put back the information we removed with our method and would require the user to fine-tune the model on the class that was removed. These classes might not be available as for example we might have removed a certain person's identity from the model whose pictures are not available on the internet and thus the class would be unrecoverable.
> >
> > **C1: Consider rewording the entire paper, for example, raise the method as a concept removal paper rather than an unlearning paper.**
> >
> > This has now been included in the revised manuscript and the title also has been revised to reflect this change.
> >
> > **C2: Consider adding an experiment to examine whether the proposed method can remove a class for several downstream datasets.**
> >
> > We have included the experiment by testing specific classes on other downstream datasets such as MS-COCO, ImageNet and also by downloading images in the wild from the web to validate the class/concept forgetting accuracy.
> >
> > **C3: Add the missing details mentioned above.**
> >
> > This has been included in the revised manuscript.
> >
> > **C4: Refine the writing according to the suggestions above.**
> >
> > The writing in the revised manuscript has been revised as per the suggestions. While most of the changes are indicated in blue, certain changes, such as changes to the citation, have not been indicated in blue but have been incorporated.

---

> > > ### Comment · Reviewer_Mfz1 · 2024-11-03
> > > **Thank you for the rebuttal**
> > >
> > > I would like to thank the authors for thoroughly revising the paper and addressing my comments. In general, I think most of my concerns are addressed. I recommend incorporating the rebuttal on W2 and W5 into the final draft.

---

> > > > ### Author Response · Authors · 2024-11-07
> > > > **Thank you!**
> > > >
> > > > Thank you very much for your feedback and we are happy that your concerns are now addressed. We will incorporate the content on points W2 and W5 from our rebuttal into the final draft!

---

### Review · Reviewer_Gerr · 2024-10-15

**Summary Of Contributions:**

This manuscript presents a zero-shot class unlearning algorithm for the vision-language model, CLIP. The proposed method employs a model adaptation technique called LoRA to disrupt the association between the visual and textual representations of the 'forget' class. It does so by adapting the projection of the textual representation into an empty representation in the image-text space while preserving the representation of the non-forget class.

**Audience:**

Yes

**Broader Impact Concerns:**

I do not have any concerns regarding the ethical implications of this work.

**Claims And Evidence:**

Yes

**Requested Changes:**

- As CLIP and LoRA are two key components of this work, I strongly encourage the authors to provide more detailed explanations. Specifically, including the loss function that CLIP optimizes and clearly formulating LoRA's adaptation strategy would be beneficial for the audience.
- I would like the authors to clarify the following points:

(1) The claims regarding forgetting: The manuscript mentions multiple times that the proposed method achieves 'complete forgetting' or 'completely removes information about the forget classes.' However, forgetting is a broad concept that cannot be easily guaranteed, and there are multiple metrics to evaluate it. For instance, in this work, forget accuracy is used to measure forgetting, but in another setting, where membership inference attacks (MIA) can be applied, the success of an MIA could be used as a measure. Clarifying this point and justifying the related claims in the manuscript will help avoid misunderstandings about the 'forgetting' performance of the unlearning algorithm.

(2) In the caption of Fig. 1, it is mentioned that 'We compare our method to five other methods, averaging across three classes for four selected datasets.' However, it is unclear which three classes the average is taken across. Does the 'Avg. Other Classes acc' refer to the average across all classes except the target class, or is it the average across the three specific classes? Additionally, I believe there is a typo in 'five other methods,' as I can only find ‘four’ other methods listed in the table.
- Equation 1 is inaccurate. Minimizing both the retain and forget loss does not reflect the optimization problem that machine unlearning addresses. Additionally, the parameter α is not defined anywhere in the manuscript. I would like the authors to provide justification for this equation.

**Strengths And Weaknesses:**

### Strengths

- The problem studied is an important real-world challenge, and providing initial solutions like the one proposed in this work is a valuable contribution.

- The research is well-structured, with insightful analysis and ablation studies.
###  Weaknesses

- A more detailed preliminary subsection on CLIP and LoRA is needed.
- Some clarifications are required
- Equation 1 needs justification

---

> ### Author Response · Authors · 2024-10-27
> **Author's Response to Reviewer Gerr**
>
> We appreciate the reviewer for their time and constructive feedback. In response to the reviewer’s suggestions, we have revised the paper accordingly and uploaded the updated manuscript with changes highlighted in blue. Below, we address the concerns raised by the reviewer.
>
> **C1: As CLIP and LoRA are two key components of this work, I strongly encourage the authors to provide more detailed explanations.**
>
> We added the detailed explanation of CLIP and LoRA introducing the text projection matrix from CLIP which is optimized with LoRA. Please see section 3 in the revised version of the paper.
>
> **C2.1: The claims regarding forgetting: The manuscript mentions multiple times that the proposed method achieves 'complete forgetting' or 'completely removes information about the forget classes.' However, forgetting is a broad concept that cannot be easily guaranteed, and there are multiple metrics to evaluate it. For instance, in this work, forget accuracy is used to measure forgetting, but in another setting, where membership inference attacks (MIA) can be applied, the success of an MIA could be used as a measure. Clarifying this point and justifying the related claims in the manuscript will help avoid misunderstandings about the 'forgetting' performance of the unlearning algorithm.**
>
> Thanks for raising this point. We have clarified this point in section 5.3 of the revised manuscript including the following:
> "It is important to note that forgetting is a broad concept that cannot be easily guaranteed and there are multiple metrics available to evaluate it. In this work, we use forget accuracy (for classification) and precision@K (for retrieval) to measure the effectiveness of forgetting. However, other metrics such as Membership Inference Attacks (MIA) exist and offer different perspectives. In our case, we cannot apply MIA because it is typically used to check if specific training data remains embedded in the model. Since we do not have access to the training data used in CLIP’s pre-training, applying MIA is not feasible in this context."
>
> **C2.2: In the caption of Fig. 1, it is mentioned that 'We compare our method to five other methods, averaging across three classes for four selected datasets.' However, it is unclear which three classes the average is taken across. Does the 'Avg. Other Classes acc' refer to the average across all classes except the target class, or is it the average across the three specific classes? Additionally, I believe there is a typo in 'five other methods,' as I can only find ‘four’ other methods listed in the table.**
>
> To clarify this better, looking at Table 10 (Appendix) for *StanfordDogs* for example, *Target Class acc.* represents the accuracy for *Pekinese, toy poodle, Scotch terrier* classes before and after forgetting. *Avg. Target Class acc.* from Table 1 (main paper) is the average across those three classes. Similarly, *Other Classes acc.* in Table 10 represents the accuracy of other classes (other classes in *StanfordDogs* for dogs example) excluding the forget class from *Class name* column and *Avg. Target Class acc.* from Table 1 for *StanfordDogs* is the average across those three classes of *Other Classes acc.* column.
>
> We also clarified what *Avg. Other Classes acc* represents in Tab. 1 in section 6.1. of the revised manuscript.
>
> Thank you for pointing out our mistake. Indeed, we have compared with four methods and not five methods. We have corrected this in the revised manuscript.
>
> **C3: Equation 1 is inaccurate. Minimizing both the retain and forget loss does not reflect the optimization problem that machine unlearning addresses. Additionally, the parameter alpha is not defined anywhere in the manuscript. I would like the authors to provide justification for this equation.**
>
> This is a valid observation. As we have now rephrased our work as class forgetting, the equation 1 that refers to the general unlearning problem is not relevant. Our work now relates specifically to class/concept forgetting and the relevant optimization problem is best described in the new equation 1 of the revised manuscript (older equation 2 of the earlier manuscript).

---

> > ### Comment · Reviewer_Gerr · 2024-11-12
> > **Rebuttal Feedback**
> >
> > Thank you to the authors for addressing my concerns in the revised version.

---

> > > ### Author Response · Authors · 2024-11-13
> > > **Thank you!**
> > >
> > > Thank you very much and we are happy that your concerns are now addressed.

---

### Review · Reviewer_HMV9 · 2024-10-18

**Summary Of Contributions:**

In this paper, the authors propose a novel method for zero-shot unlearning in CLIP. The proposed method targets the text to image projection matrix, rather than both image and text, and uses LoRA + finetuning to cause a textual class a practitioner wishes CLIP unlearn to project further away from corresponding images. By doing so, the authors emphasize that exact image/text pairs are not required for unlearning, which is helpful in the context of large models where original datasets are not readily available. Extensive experimental results across several datasets show that the authors’ proposed method generalizes well across both ViT and CNN architectures, and precisely targets specific class(es) for unlearning while preserving performance on other classes. Additionally, the authors investigate which neurons are being affected the most during the unlearning process, and find that there are certain “magic neurons” that have the greatest absolute change.

**Audience:**

Yes

**Claims And Evidence:**

Yes

**Requested Changes:**

#

- [critical] given that $A_{\{ds\}}$ represents all the classes in the dataset minus the unlearned class, can the authors justify that this is a representative sample of classes whose performance should be unaffected? Is it possible for the authors to separately evaluate accuracy of unaffected classes from a standardized dataset (mscoco)?
- [critical] can the authors report on the accuracy of the 100 semantically similar classes generated during the unlearning procedure? Is the performance simlar to the “Avg. Other Class acc.” from the table?
- [critical] since LoRA is used to perform the unlearning, have the authors also attempted sequential unlearning of classes? I.e. where requests to unlearn data does not arrive all at once, but over different time periods. Is the proposed method still effective?
- [clarifying] the work in section 6.3 suggests that across different forgetting concepts, it appears the same “magic” neuron is being targeted the most during unlearning. That seems to suggest that unlearning should have a large effect on all input images regardless of what class they are. Can the authors provide clarification/insight into why this is not the case?
- [clarifying] in the context of model forgetting, the goal is to *completely* remove the effect that certain data have had on an ML model. However even after running the unlearning procedure proposed in this paper, the projected space in which an image corresponding to the target class still exists and has not been removed. Thus unlearning can not be entirely achieved, as the effect of certain data points still exists in the model. Can the authors clarify this in the manuscript?

**Strengths And Weaknesses:**

**Strengths**

- proposed method significantly outperforms existing zero-shot forgetting in CLIP models
- generalizes well across both CNN and ViT based CLIP architectures

**Weaknesses**

- the authors motivate unlearning in the context of ensuring a user’s data can be entirely removed from models, however the proposed technique doesn’t exactly fit this problem, and seems to address more along the topic of “concept forgetting”
- forgetting a concept will lead to degradation of neighbor concepts, which is not quantifiably measured in experiments. This makes it harder for model practitioners to accept the proposed method due to an undefined tradeoff

---

> ### Author Response · Authors · 2024-10-27
> **Author's Response to Reviewer HMV9 - Part 1**
>
> We appreciate the reviewer for their time and constructive feedback. In response to the reviewer’s suggestions we have revised the paper accordingly and uploaded the updated manuscript with changes highlighted in blue. Below, we address the concerns raised by the reviewer.
>
> **W1: The authors motivate unlearning in the context of ensuring a user’s data can be entirely removed from models; however, the proposed technique doesn’t exactly fit this problem and seems to address “concept forgetting” more.**
>
> We agree with the reviewer and we have rephrased our paper as a class/concept forgetting rather than an unlearning work in the revised manuscript. Specifically, please refer to the revised title and introduction section in blue for the main changes.
>
> **W2: Forgetting a concept may lead to degradation of neighbor concepts, which is not quantifiably measured in experiments. This makes it harder for model practitioners to accept the proposed method due to an undefined tradeoff.**
>
> Thanks for raising this point. We were specifically interested in validating this and respond to this in detail in the clarification points below.
>
> **C1: Given that $A_{ds}$ represents all classes in the dataset minus the unlearned class, can the authors justify that this is a representative sample of classes whose performance should remain unaffected? Is it possible to separately evaluate accuracy of unaffected classes from a standardized dataset (MSCOCO)?**
>
> Please note that in our paper $A_{ds}$ is the dataset, not the classes. We added some additional details into section 5.3 to make things more clear. In terms of representative sample of classes, we can identify two general groups: semantically similar and non-semantically similar classes. We find in our experiments that semantically similar classes are always the ones most affected while the impact of class forgetting on non-semantically similar classes is often very small. We provide a thorough evaluation of both groups in our experiments, thus we believe that we in general include a representative number of classes. Below we discuss this in more details.
>
> Let us clarify this point starting with summarizing how we evaluate class forgetting looking at the table below (named *Forgetting results with RN50 visual encoder.*) which is a row taken from Table 10 (Appendix) that provides granular results related to the aggregated results in Table 1 of the main paper. Assume we want to forget "Pekinese" from *StanfordDogs*. To forget "Pekinese" class we generate, using ChatGPT, semantically similar classes of "Pekinese" and the output are different breeds of dogs (note that these are normally different from the actual *StanfordDogs* classes but there will be some overlapping classes). The column labeled *Target Class acc.* (BF and AF) represents the remaining accuracy for "Pekinese" class before and after forgetting which goes to 0 in AF column meaning complete forgetting. The column *Other Classes acc* measures the model's accuracy on the remaining classes from *StanfordDogs* that are semantically similar as they are also dog breeds. These results are shown before and after forgetting and are almost the same meaning that other **semantically similar** dog classes were unaffected after forgetting "Pekinese". This addresses one of the key concerns raised in the review, which mentioned that *forgetting a concept could degrade neighboring concepts and this has not been quantitatively measured*, our method indeed captures and quantifies this effect as shown by the performance on semantically similar classes that we believe can be considered *neighboring concepts*.
>
> Additionally, the accuracy on **semantically different** classes, such as those from the *StanfordCars*, *Caltech101*, and *OxfordFlowers* datasets, remains stable. This demonstrates that both semantically similar and semantically different classes are preserved after forgetting the "Pekinese" class. Even within *Caltech101*, which contains some potentially semantically similar classes (e.g., some animal classes), the unchanged accuracy after unlearning further confirms that these classes are unaffected.
>
>
> **Dataset**     | **Class name** | **Target Class acc. BF** | **Target Class acc. AF** | **Other Classes acc. BF** | **Other Classes acc. AF** | **StCars BF**  | **StCars AF** | **StDogs BF**  | **StDogs AF** | **Cal101 BF**  | **Cal101 AF** | **Flowers BF**  | **Flowers AF** |
> |-----------------|----------------|--------------------------|----|---------------------------|----|----------------------|----|----------------------|----|---------------------|----|-----------------------|----|
> StDogs    | Pekinese       | 0.705                    | 0.0| 0.515                     | 0.51| 0.558               | 0.559| -                  | -  | 0.857               | 0.853| 0.661                 | 0.659 |
>
> **Table:** Forgetting results with RN50 visual encoder.

---

> ### Author Response · Authors · 2024-10-27
> **Author's Response to Reviewer HMV9 - Part 2**
>
> Importantly, Table 6 in the main paper (row *SemDiff*) shows that even if we don't use semantically similar classes to the forget class as the retain classes during the optimization procedure, in our example using classes from Food101 dataset, the accuracy on semantically similar classes to the forget class drops only by approximately 3-4%, while accuracy on non-semantically similar classes remains unchanged. This reinforces that the forgetting process primarily affects semantically related classes.
>
> As requested by the reviewer we also evaluate the not unlearned RN50 CLIP model and CLIP model with "Scotch terrier" class unlearned on MS-Coco and ImageNet datasets using validation set. These results are shown in the Table below where we can see that the accuracy of unaffected classes on standardized datasets remains stable after forgetting.
>
> | **Dataset**           | **Unlearned CLIP** | **Not Unlearned CLIP** |
> |-----------------------|--------------------|-------------------------|
> | MS-COCO (all classes) | 0.9661            | 0.9667                 |
> | ImageNet (all classes)| 0.5756            | 0.5794                 |
>
> **C2: Can the authors report on the accuracy of the 100 semantically similar classes generated during the unlearning procedure? Is the performance similar to the “Avg. Other Class acc.” from the table?**
>
> In Table 6 of the main paper (we show part of it in the table below for easier navigation) we report the results of class forgetting where the actual classes from the dataset are used to retain the information about the non-forget classes (row $Cls_r$). In other words, taking *StanfordDogs* dataset that contains 120 classes overall and we want to forget "Pekinese", we use the other 119 textual classes from *StanfordDogs* to retain the information about other classes. As we can see in the table below in column *Avg. Other Classes acc* BF/AF the accuracy on the semantically similar classes remains almost unchanged. This is the proxy for the experiment that the reviewer is asking as we do not have the images for all the textual semantically similar classes generated by ChatGPT.
>
>
>
> | **Type of Retained Classes** | **Model** | **Avg. Target Class acc. BF** | **Avg. Target Class acc. AF** | **Avg. Other Classes acc. BF** | **Avg. Other Classes acc. AF** | **Avg. StCars BF** | **Avg. StCars AF** | **Avg. StDogs BF** | **Avg. StDogs AF** | **Avg. Cal101 BF** | **Avg. Cal101 AF** | **Avg. Flowers BF** | **Avg. Flowers (AF)** | **Avg. Score (↓)** |
> |------------------------------|-----------|-------------------------------|--------|--------------------------------|--------|---------------------------|--------|--------------------------|--------|-------------------------|--------|---------------------------|--------|---------------------|
> | $Cls_{r}$                    | RN50      | 0.669                         | 0.0    | 0.648                          | 0.644  | 0.558                     | 0.558  | 0.517                    | 0.51   | 0.857                   | 0.856  | 0.661                     | 0.659  | **0.004**            |
> | *SemSim*                     | RN50      | 0.669                         | 0.0    | 0.648                          | 0.642  | 0.558                     | 0.557  | 0.517                    | 0.513  | 0.857                   | 0.857  | 0.661                     | 0.656  | 0.005               |
> | *SemDiff*                    | RN50      | 0.669                         | 0.0    | 0.648                          | 0.61   | 0.558                     | 0.558  | 0.517                    | 0.511  | 0.857                   | 0.86   | 0.661                     | 0.655  | 0.016               |

---

> > ### Author Response · Authors · 2024-10-27
> > **Author's Response to Reviewer HMV9 - Part 3**
> >
> > **C3: since LoRA is used to perform the unlearning, have the authors also attempted sequential unlearning of classes? I.e. where requests to unlearn data does not arrive all at once, but over different time periods. Is the proposed method still effective?**
> >
> > For sequential class forgetting we can see the results in the table below. We first forget the first class (in *Classes* column), e.g. "Pekinese" in *StanfordDogs* and then forget the other two classes ("toy poodle","Scotch terrier") starting from the model that has forgotten the first class. As we can see,  forgetting still works well even in sequential forgetting when forgetting requests arrive at different times. The results of this sequential forgetting are similar to multiple class forgetting in the main paper.
> >
> > | **Model** | **Dataset** | **Classes** | **Avg. Target Classes acc. (BF)** | **Avg. Target Classes acc. (AF)** | **Other Classes acc. (BF)** | **Other Classes acc. (AF)** | **StCars (BF)** | **StCars (AF)** | **StDogs (BF)** | **StDogs (AF)** | **Cal101 (BF)** | **Cal101 (AF)** | **Flowers (BF)** | **Flowers (AF)** | **Avg. Score (↓)** |
> > |-----------|-------------|-------------|-----------------------------------|--------|-----------------------------|--------|------------------------|--------|------------------------|--------|-----------------------|--------|------------------------|--------|-----------------------|
> > | RN50      | StDogs | Pekinese, toy poodle, Scotch terrier | 0.591 | 0.011 | 0.515 | 0.498 | 0.558 | 0.542 | - | - | 0.857 | 0.854 | 0.661 | 0.633 | 0.025 |
> > | RN50      | StCars | 2009 Spyker C8 Coupe, 2010 Dodge Ram Pickup 3500 Crew Cab, 2011 Ford Ranger SuperCab | 0.397 | 0.0 | 0.56 | 0.528 | - | - | 0.517 | 0.499 | 0.857 | 0.847 | 0.661 | 0.653 | 0.023 |
> > | RN50      | Cal101  | euphonium, minaret, platypus | 0.827 | 0.0 | 0.858 | 0.864 | 0.558 | 0.55 | 0.517 | 0.498 | - | - | 0.661 | 0.656 | 0.012 |
> > | RN50      | Flowers | trumpet creeper, gazania, tree mallow | 0.86 | 0.0 | 0.656 | 0.626 | 0.558 | 0.555 | 0.517 | 0.502 | 0.857 | 0.857 | - | - | 0.016 |
> >
> > **C4: the work in section 6.3 suggests that across different forgetting concepts, it appears the same “magic” neuron is being targeted the most during unlearning. That seems to suggest that unlearning should have a large effect on all input images regardless of what class they are. Can the authors provide clarification/insight into why this is not the case?**
> >
> > Please note that the "magic" neuron is not a single neuron but consists of all the neurons in column 222 of the projection matrix, which contains 512 values for the RN50 model case. During the forgetting process, using LoRA adaptation, the projection matrix is adjusted primarily by modifying column 222 and its 512 values. From our analysis, this modification significantly impacts feature 222 only when the textual embedding of the class to be forgotten passes through it. For example, if the class to be forgotten is "Scotch Terrier," feature 222 undergoes substantial changes after passing through the modified text projection matrix, increasing the value of feature 222 significantly so that it's pushed towards 0 and the dot product with the image embedding for that class becomes small. However, for other classes, feature 222 remains relatively unchanged, so the dot product remains high, allowing these classes to still be accurately classified. In table below we observe that the value of feature 222 changes significantly for "Scotch Terrier" between the unlearned and not unlearned models, while it remains stable for other classes ("Toy Poodle", "Pekinese", "Spaniel"). This ensures that the unlearning process only affects the targeted class without disrupting the other classes.
> >
> > | **Class**                    | **Unlearned Model** | **Not Unlearned Model** |
> > |---------------------------------|---------------------|-------------------------|
> > | Scotch Terrier (Forgotten class) | -0.9342            | -4.1981                 |
> > | Toy Poodle                       | -6.5257            | -6.5185                 |
> > | Pekinese                         | -4.3343            | -3.9811                 |
> > | Spaniel                          | -6.5226            | -6.8187                 |

---

> > > ### Author Response · Authors · 2024-10-27
> > > **Author's Response to Reviewer HMV9 - Part 4**
> > >
> > > **C5: in the context of model forgetting, the goal is to completely remove the effect that certain data have had on an ML model. However, even after running the unlearning procedure proposed in this paper, the projected space in which an image corresponding to the target class still exists and has not been removed. Thus, unlearning cannot be entirely achieved, as the effect of certain data points still exists in the model. Can the authors clarify this in the manuscript?**
> > >
> > > This observation is correct. Since we do not alter the visual encoder, the effect of the classes to forget still exists there. Hence, tasks such as image retrieval from an image input, which rely on visual similarity, can still succeed. However, we argue that while the uni-modal information remains in the model, breaking the multi-modal association (between text and image) is sufficient *specifically* for class forgetting. This is because during image retrieval from an image input, a strong model like CLIP can still identify similar features and shapes of objects *without actually recognizing or knowing the specific textual class they belong to*. This can be seen when performing image-to-image retrieval using the **original** CLIP model on classes where its classification accuracy is **zero**. The results presented in the table below support this. Therefore, we conclude that breaking the text-image link is sufficient for achieving class forgetting for CLIP.
> > >
> > >
> > > | **Model Type**           | **Class**                                           | **Classification Accuracy** | **Precision@1** | **Precision@5** | **Precision@10** |
> > > |--------------------------|-----------------------------------------------------|------------------------------|------------------|------------------|-------------------|
> > > | CLIP original            | Appenzeller (StanfordDogs)                         | 0                            | 1.0              | 0.4              | 0.2               |
> > > | CLIP original            | Pembroke (StanfordDogs)                            | 0                            | 1.0              | 0.6              | 0.4               |
> > > | CLIP original            | Cardigan (StanfordDogs)                            | 0                            | 0.0              | 0.2              | 0.2               |
> > > | CLIP original            | 2010 Chevrolet HHR SS (StanfordCars)               | 0                            | 1.0              | 0.4              | 0.4               |
> > > | CLIP original            | 2009 HUMMER H2 SUT Crew Cab (StanfordCars)        | 0                            | 1.0              | 0.6              | 0.7               |
> > > | CLIP original            | english marigold (OxfordFlowers)                   | 0                            | 1.0              | 0.8              | 0.6               |
> > > | CLIP original            | colt's foot (OxfordFlowers)                        | 0                            | 1.0              | 0.8              | 0.7               |
> > > | CLIP original            | cape flower (OxfordFlowers)                         | 0                            | 1.0              | 1.0              | 1.0               |

---

> > > > ### Comment · Reviewer_HMV9 · 2024-11-12
> > > > **Thank you for rebuttal**
> > > >
> > > > I would like to thank the authors for addressing my comments with thorough experimentation. Please include W2 in the final manuscript.

---

> > > > > ### Author Response · Authors · 2024-11-13
> > > > > **Thank you!**
> > > > >
> > > > > Thank you very much. We will incorporate W2 into the final manuscript!

---

### Decision · Action_Editor_isyL · 2024-12-03

**Recommendation:** Accept with minor revision

**Comment:**

The paper looks basically acceptable, but I am recommending "accept with minor revision" to make sure the authors add the new results they have promised in the comments.

**Audience:**

All reviewers agree that the paper makes an interesting contribution to the topic of class forgetting in CLIP models and would be interesting to readers interested in this problem.

**Claims And Evidence:**

All reviewers agree that the submission is supported by accurate, convincing and clear evidence.

---

> ### Author Response · Authors · 2024-12-23
> **Thanks!**
>
> We have incorporated the revision points outlined in the "Main Changes" section of the "Phase One Response" into the final manuscript. To ensure the final version stays within the 12-page limit, we have moved the "Retrieval Task" to Appendix C.
> As requested by Mfz1, we have added W2 and W5 to Appendix E. Additionally, per HMV9’s feedback, we clarified W2-C1 in Section 6.1, emphasizing that we evaluate the remaining accuracy on both semantically similar and semantically different classes after forgetting. We have also included experiments on MS-COCO and ImageNet in Appendix I.
> Further, W2-C3 has been added to Appendix G, and W2-C4 to Appendix H. Lastly, we have introduced a new experiment on Identity Forgetting in Section 6.2, demonstrating that our method can effectively forget faces.
> We have also added the link to the github page in the paper and uploaded the camera-ready supplementary video.
> We thank everyone for a smooth and swift review process